# Transolver++: An Accurate Neural Solver for PDEs on Million-Scale Geometries

**Huakun Luo** [* 1]   **Haixu Wu** [* 1]   **Hang Zhou** [1]   **Lanxiang Xing** [1]   **Yichen Di** [1]   **Jianmin Wang** [1]   **Mingsheng Long** [1]

## Abstract

Although deep models have been widely explored in solving partial differential equations (PDEs), previous works are primarily limited to data only with up to tens of thousands of mesh points, far from the million-point scale required by industrial simulations that involve complex geometries. In the spirit of advancing neural PDE solvers to real industrial applications, we present Transolver++, a highly parallel and efficient neural solver that can accurately solve PDEs on million-scale geometries. Building upon previous advancements in solving PDEs by learning physical states via Transolver, Transolver++ is further equipped with an extremely optimized parallelism framework and a local adaptive mechanism to efficiently capture eidetic physical states from massive mesh points, successfully tackling the thorny challenges in computation and physics learning when scaling up input mesh size. Transolver++ increases the single-GPU input capacity to million-scale points for the first time and is capable of continuously scaling input size in linear complexity by increasing GPUs. Experimentally, Transolver++ yields 13% relative promotion across six standard PDE benchmarks and achieves over 20% performance gain in million-scale high-fidelity industrial simulations, whose sizes are $100\times$ larger than previous benchmarks, covering car and 3D aircraft designs. Code is available at https://github.com/thuml/Transolver_plus.

## 1. Introduction

Extensive physics processes can be precisely described by partial differential equations (PDEs) (Wazwaz, 2002; Evans, 2010), such as air dynamics of driving cars or internal stress of buildings. Accurately solving these PDEs is essential to

---

[*]Equal contribution  [1]School of Software, BNRist, Tsinghua University. Huakun Luo <luohk24@mails.tsinghua.edu.cn>. Correspondence to:Mingsheng Long<mingsheng@tsinghua.edu.cn>.

*Proceedings of the $42^{nd}$ International Conference on Machine Learning*, Vancouver, Canada. PMLR 267, 2025. Copyright 2025 by the author(s).

industrial manufacturing (Kopriva, 2009; Roubíček, 2013). However, it is hard and usually impossible to obtain the analytic solution of PDEs. Thus, numerical methods (Ŝolín, 2005) have been widely explored, whose typical solving process is first discretizing PDEs into computation meshes and then approximating solution on discretized meshes (Grossmann et al., 2007). In real applications, such as simulating a driving car or aircraft during takeoff, it will take several days or even months for calculation, and the simulation accuracy is highly affected by the fitness of discretized computation mesh (Solanki et al., 2003; Elrefaie et al., 2024). To speed up this process, deep models have been explored as efficient surrogates of numerical methods, known as neural PDE solvers (Wang et al., 2023). Training from pre-collected simulation data, neural solvers can learn to approximate the mapping between input and output of numerical methods and directly infer new samples in a flash (Li et al., 2021), posing a promising direction for industrial simulation.

Usually, industrial applications involve large and complex geometries, requiring the model to capture intricate physics processes underlying geometries efficiently. Although previous works have made some progress in handling complex geometries (Li et al., 2023b; Hao et al., 2023) and attempted to speed up model efficiency, they are still far from real applications, especially in handling large geometries. Specifically, as illustrated in Figure 1(a), existing models fail to scale up beyond 400k points, while real applications typically involves million-scale mesh points or even more. Note that, as aforementioned, the fineness of computation meshes is the foundation of simulation accuracy. As the comparison of car mesh shown in Figure 1(b), limited mesh size will seriously sacrifice the precision of the geometry, where the originally streamlined surface becomes rough and uneven. This will further limit the simulation accuracy of physics processes, especially for aerodynamic applications (Elrefaie et al., 2024). Thus, *the capability of handling large geometries is indispensable for practical neural PDE solvers.*

As the latest progress in neural PDE solver architectures, Transolver (Wu et al., 2024) proposes to learn intrinsic physical states underlying complex geometries and apply the attention mechanism among learned physical states to capture intricate physics interactions, which frees model capacities from unwieldy mesh points and achieves outstanding performance in car and airfoil simulations. Although Transolver

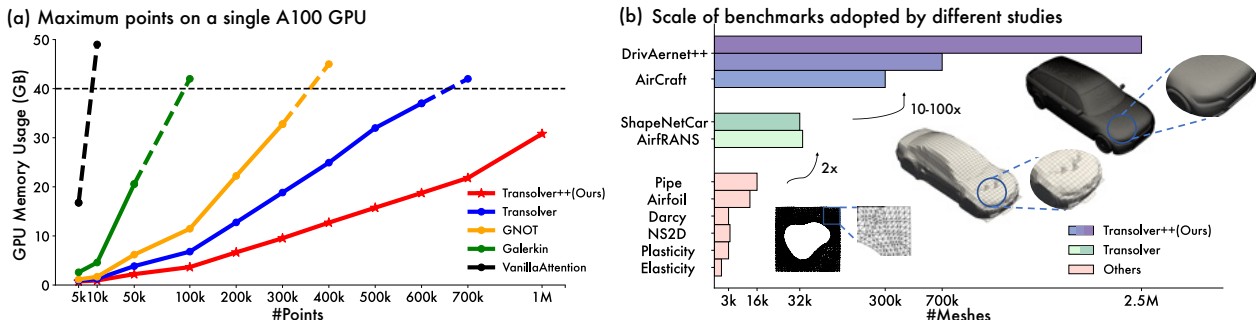

*Figure 1.* (a) Comparison of model capability in handling large geometries. We plot the GPU memory change of each model when increasing input mesh points. The upper bound on a single A100 40GB GPU is depicted in the dotted line. (b) Comparison of experiment benchmarks. Transolver++ experiments on high-fidelity tasks with up to 2.5 million points, which is $100\times$ larger than previous works.

demonstrates favorable capability in handling complex geometries, its maximum input size remains limited to 700k points (Figure 1(a)), and its experiment data is much simpler than real applications. Building on the essential step made by Transolver, we attempt to advance it to million-scale or even larger geometries in pursuing practical neural solvers.

When scaling Transolver to million-scale high-fidelity PDE-solving tasks, we observe its bottlenecks in physics learning and computation efficiency. Firstly, as Transolver highly relies on the learned physical states, massive mesh points may overwhelm their learning process, resulting in homogeneous physical states and model degeneration. Secondly, it is observed that even deep representations of million-scale points without considering intermediate calculations will consume considerable GPU memory, which is strictly constrained by total resources on a single GPU. This drives us to elaboratively optimize the model architecture and unlock the power of multi-GPU parallelism. Thus, in this paper, we present *Transolver++*, which upgrades Transolver with an extremely optimized parallelism framework and a local adaptive mechanism to efficiently capture eidetic physical states from massive mesh points. Based on the co-design with model architecture, our parallelism framework significantly reduces the communication overhead and achieves linear scalability with resources without sacrificing performance. As a result, Transolver++ achieves consistent state-of-the-art on six standard PDE datasets and successfully extends Transolver to high-fidelity car and aircraft simulation tasks with million-scale points. Here are our contributions:

- To ensure reliable modeling for complex physical interactions, we introduce Transolver++ with eidetic states, which can adaptively aggregate information from massive mesh points to distinguishable physical states.

- We present an efficient and highly parallel implementation of Transolver++ with linear scalability and minimal overhead across GPUs, affording mesh size of 1.2 million on a single GPU while maintaining accuracy.

- Transolver++ achieves a 13% relative gain averaged

from six standard benchmarks and over 20% improvement on high-fidelity million-scale industrial datasets, covering practical car and 3D aircraft design tasks.

## 2. Related Work

### 2.1. Neural PDE Solver

Traditional numerical methods for solving PDEs often require high computational costs to achieve accurate solutions (Solanki et al., 2003). Recently, deep learning methods have demonstrated remarkable potential as efficient surrogate models for solving PDEs due to their inherent nonlinear modeling capability, known as neural PDE solvers.

As a typical paradigm, operator learning has been widely studied for solving PDEs by learning the mapping between input functions and solutions. FNO (Li et al., 2020a) first proposes to approximate integral in the Fourier domain for PDE solving. Subsequently, Geo-FNO (Li et al., 2021) extends FNO to irregular meshes by transforming them into regular grids in the latent space. To further enhance the capabilities of FNO, U-FNO Wen et al. (2022) and U-NO Rahman et al. (2023) are presented by leveraging U-Net (Ronneberger et al., 2015) to capture multiscale properties. Considering the high dimensionality of real-world PDEs, LSM (Wu et al., 2023) applies spectral methods in a learned lower-dimensional latent space to approximate input-output mappings. Afterward, LNO (Wang & Wang, 2024) adopts the attention mechanism to effectively map data from geometric space to latent space for complex geometries.

Recently, Transformers (Vaswani et al., 2017) have achieved impressive progress in extensive fields and have also been applied to solving PDEs, where the attention mechanism has been proven as a Monte-Carlo approximation for global integral (Kovachki et al., 2023). However, standard attention suffers from quadratic complexity. Thus, many models like Galerkin (Cao, 2021), OFormer (Li et al., 2023c) and FactFormer (Tran et al., 2023) propose different efficient attention mechanisms. Among them, GNOT (Hao

et al., 2023) utilizes well-established linear attention, like Reformer or Performer (Kitaev et al., 2020; Choromanski et al., 2021), and separately encodes geometric information, achieving favorable performance. However, linear attention often suffers from degraded performance as an approximation of standard attention (Qin et al., 2022). Moreover, these Transformer-based methods treat input geometries as a sequence of mesh points and directly apply attention among mesh points, which may fall short in geometric learning and computation efficiency. As a significant advancement in PDE solving, Transolver (Wu et al., 2024) introduces the Physics-Attention mechanism that groups massive mesh points into multiple physical states and applies attention among states, thereby enabling more effective and intrinsic modeling of complex physical correlations. However, it still faces challenges in degenerated physics learning and a high computation burden under million-scale geometries. These challenges will be well addressed in our paper.

In addition to Transformers, graph neural networks (GNNs) (Hamilton et al., 2017; Gao & Ji, 2019; Pfaff et al., 2021) are also inherently suitable to process unstructured meshes by explicitly modeling the message passing among nodes and edges. GNO (Li et al., 2020b) first implements neural operator with GNN. Later, GINO (Li et al., 2023a) combines GNO with Geo-FNO to encode the geometry information. Recently, 3D-GeoCA (Deng et al., 2024) integrates pretrained 3D vision models to achieve a better representation learning of geometries. However, prone to geometric instability (Klabunde & Lemmerich, 2023), GNNs can lead to unstable results and may be insufficient in capturing global physics interactions, especially when handling large-scale unstructured meshes (Morris et al., 2023).

### 2.2. Parallelism of Deep Models

Despite processing large-scale geometries being crucial in industrial design, this problem has not been explored in previous research. This paper breaks the computation bottleneck by leveraging the parallelism framework, which is related to the following seminal works. Towards general needs in handling the high GPU memory usage caused by large-scale inputs, several parallel frameworks have been proposed, such as tensor parallelism (Shoeybi et al., 2019) and model parallelism (Huang et al., 2019). However, these methods are highly model-dependent and request significant communication overhead (Zhuang et al., 2023). Another direction is to optimize attention mechanisms. Ring attention (Liu et al., 2023), inspired by FlashAttention (Dao et al., 2022), uses a ring topology between multi-GPUs, achieving quadratic communication complexity with respect to mesh points. Besides, DeepSpeed-Ulysses (Jacobs et al., 2023) splits the deep representation along the channel dimension and employs the All2All communication approach, reducing complexity to linear. Despite these improvements,

the communication volume remains excessive. In contrast, Transolver++ leverages its unique physics-learning design and presents a highly optimized parallelism framework tailored to PDE solving, enabling minimal communication overhead and allowing for meshes of million-scale points.

## 3. Revisiting Transolver

Before Transolver++, we would like to briefly introduce Physics-Attention, the key design of Transolver (Wu et al., 2024), and discuss its scaling issues for million-scale inputs.

Given meshes with $N$ nodes, to capture intrinsic physics interactions under unwieldy mesh points, Physics-Attention will first assign $C$-channel input points $\mathbf{x} = \{\mathbf{x}_i\}_{i=1}^N \in \mathbb{R}^{N \times C}$ to $M$ *physical states* $\mathbf{s} = \{\mathbf{s}_j\}_{j=1}^M \in \mathbb{R}^{M \times C}$ based on the slice weights $\mathbf{w} = \{\mathbf{w}_i\}_{i=1}^N \in \mathbb{R}^{N \times M}$ learned from inputs, and each $\mathbf{w}_i \in \mathbb{R}^{1 \times M}$ represents the possibility that $\mathbf{x}_i$ belongs to each state. Specifically, physical states are aggregated from all point representations based on learned slice weights, which can be formalized as:

$$\text{Slice weights: } \mathbf{w} = \text{Softmax}\left(\text{Linear}(\mathbf{x})/\tau_0\right)$$

$$\text{Physical states: } \{\mathbf{s}_j\}_{j=1}^M = \left\{\frac{\sum_{i=1}^N \mathbf{w}_{ij}\mathbf{x}_i}{\sum_{i=1}^N \mathbf{w}_{ij}}\right\}_{j=1}^M, \quad (1)$$

where $\tau_0$ is the temperature constant. Next, the canonical attention mechanism is applied to learned physical states to capture underlying physics interactions:

$$\mathbf{q}, \mathbf{k}, \mathbf{v} = \text{Linear}(\mathbf{s}), \ \mathbf{s}' = \text{Softmax}\left(\frac{\mathbf{q}\mathbf{k}^\top}{\sqrt{C}}\right)\mathbf{v}. \quad (2)$$

Finally, Physics-Attention employs the *deslice* operation to map updated states $\mathbf{s}'$ back to mesh space using slice weights, i.e. $\mathbf{x}' = \{\mathbf{x}_i'\}_{i=1}^N = \{\sum_{j=1}^M \mathbf{w}_{ij}\mathbf{s}_j'\}_{i=1}^N$. By replacing standard attention in Transformer (Vaswani et al., 2017) with Physics-Attention, we can obtain the Transolver.

Although Transolver successfully reduces canonical computation complexity from $\mathcal{O}(N^2)$ to $\mathcal{O}(M^2)$ by learning physical states ($M$ is a constant and usually set as 32 or 64 in practice), Transolver still face the following challenges when scaling input size to million-scale, i.e. $N \geq 10^6$.

**Homogeneous physical states**  Eq. (1) shows that physical states are highly affected by slice weights $\mathbf{w}$. If $\mathbf{w}$ tends to be uniform, attention in Eq. (2) will degenerate to average pooling, losing the physics modeling capability. As shown in Figure 2(b), we find that Transolver may generate less distinguishable weights in some cases, especially in large-scale meshes, leading to homogeneous physical states.

**Efficiency bottleneck**  Although Physics-Attention cost is nearly constant when scaling the input, the feedforward layer to embed million-scale points by $\mathbf{x}_i$ will consume huge GPU memory, forming Transolver's stability bottleneck.

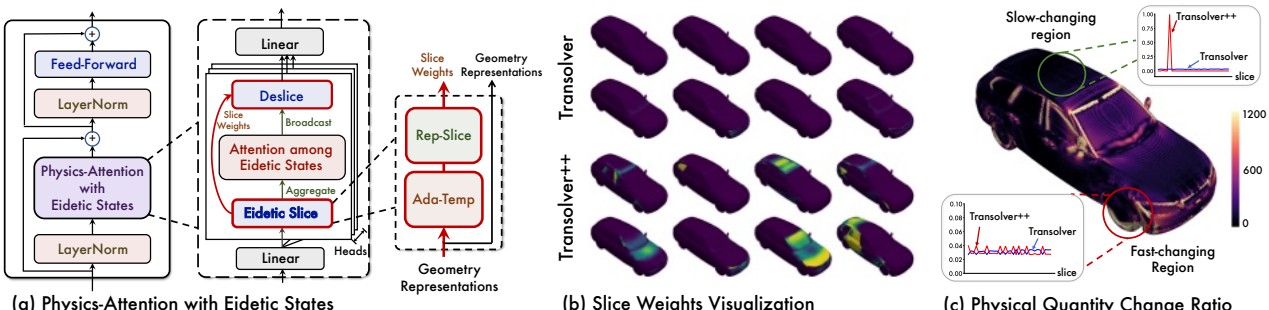

*Figure 2.* (a) Overall design of Transolver++ block. Blocks highlighted in red represent modifications compared to the original Transolver. (b) Visualization of slice weights. Transolver++ learns more diverse and eidetic physical states. (c) Visualizations of physical quantity change ratio (difference between each point and its neighbors) and slice weights learned by models. The lighter color means faster change.

## 4. Transolver++

To tackle the scaling issues of Transolver in physics learning and computation efficiency, we present Transolver++, which can effectively avoid attention degeneration by learning eidetic physical states and successfully break the efficiency bottleneck with a highly optimized parallelism framework.

### 4.1. Physics-Attention with Eidetic States

As aforementioned, if Physics-Attention is based on homogeneous physical states, it will degenerate to average pooling, which will damage the model's performance. Thus, as the authors mentioned in Transolver (Wu et al., 2024), they adopt the softmax function in calculating slice weights in Eq. (1), which can alleviate the indistinguishable states to some extent by guaranteeing peakier distribution (Wu et al., 2022). However, we still observe the degeneration phenomenon when the model depth increases, as shown in Figure 2(b), which may result from the excessive focus on global information. Note that in industrial applications, the model's capability of learning subtle physics is essential, which may highly affect the evaluation in industrial design (Elrefaie et al., 2024), as a small diversion device can drastically change the wind drag of driving cars.

To capture detailed physics phenomena, we propose to learn eidetic states by augmenting Transolver's state learning with a local-adaptive mechanism and slice reparameterization to carefully control the learned slice weight distribution.

**Local adaptive mechanism** As shown in Figure 2(b), modeling the distribution of each mesh point using a non-parametric approach has been proven to be impractical (Ye et al., 2024). Thus, we introduce a local adaptive mechanism that incorporates local information as a pointwise adjustment to the physics learning process. Specifically, we change the sharpness of the state distribution by learning to adjust the temperature $\tau_0$ in the Softmax function, namely

$$\text{Ada-Temp: } \tau = \{\tau_i\}_{i=1}^N = \{\tau_0 + \text{Linear}(\mathbf{x}_i)\}_{i=1}^N, \quad (3)$$

where $\tau \in \mathbb{R}^{N \times 1}$ and a higher temperature forms a more uniform distribution, while a lower temperature makes the

distribution more concentrated on crucial states. Through a learnable linear projection layer, we can dynamically adjust the state distribution based on each point's local properties.

**Slice reparameterization** As mentioned above, we try to learn eidetic physical states and assign mesh points to physical states with a possibility $\mathbf{w}$. In the canonical design of Transolver, it uses $\text{Softmax}$ to form a categorical distribution across states. However, simply generating a categorical distribution is not enough, as we have not completely modeled the assignment process from points to certain physical states. Considering that direct sampling via $\text{Argmax}$ is non-differentiable, we propose to adopt the Gumbel-Softmax (Jang et al., 2017) to perform differentiable sampling from the discrete categorical distribution, which is accomplished with the following reparameterization technique:

$$\text{Rep-Slice}(\mathbf{x}, \tau) = \text{Softmax}\left(\frac{\text{Linear}(\mathbf{x}) - \log(-\log \epsilon)}{\tau}\right), \quad (4)$$

where $\tau \in \mathbb{R}^{N \times 1}$ is the local adaptive temperature and $\epsilon = \{\epsilon_i\}_{i=1}^N, \epsilon_i \sim \mathcal{U}(0, 1)$. Here $\log(-\log \epsilon_i) \sim \text{Gumbel}(0, 1)$, where $\text{Gumbel}$ is a type of generalized extreme value distribution. Replacing Transolver's slice weights $\mathbf{w}$ in Eq. (1) by our new design in Eq. (4), Transolver++ is able to learn eidetic physical states under complex geometries, offering the possibility to handle much larger-scale datasets.

As shown in Figure 2(c), the slice weights in Transolver++ can perfectly adapt to the intricate physics fields on complex geometries. Specifically, regions with slow-changing physics quantities are assigned to one certain eidetic state as these areas are governed by one pure physical state. In contrast, regions with fast-changing physics quantities exhibit a multimode distribution across physics states, reflecting that these areas are influenced by a mixture of multiple states.

### 4.2. Parallel Transolver++

To break the GPU memory bottleneck caused by feedforward layers of million-scale point representations, we design

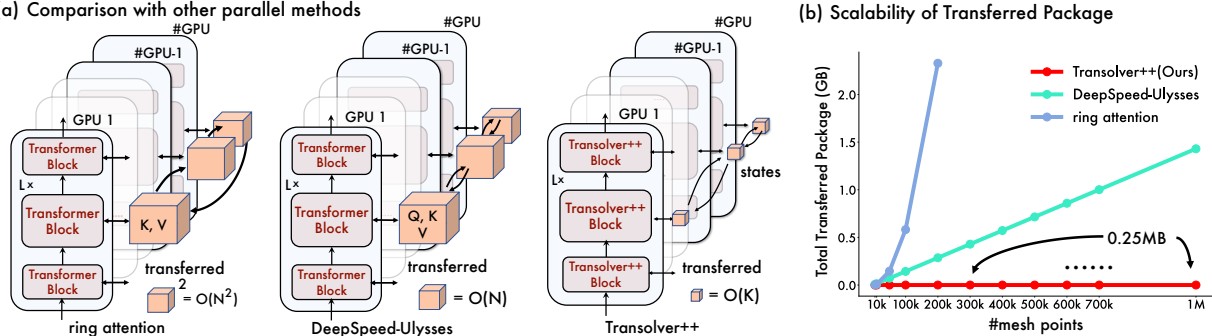

*Figure 3.* (a) Comparison with other parallel methods. Tailored to the unique physics learning design, our method only communicates physical states with an all-reduce operation. (b) Scalability of the communication overhead to the number of mesh points with 32 GPUs. Our parallel method stands out by only transferring 0.25MB of data, which does not scale with the size of input mesh points.

a highly optimized framework for PDE solving, which is based on the unique physics-learning design of Transolver.

**Parallelism formulation**  Through careful analysis, we observe that in addition to the point-wise feedforward layer, the computation of eidetic states within Physics-Attention can be efficiently distributed across multiple GPUs. Specifically, operations in Eq. (1), such as the weighted sum $\sum_{i=1}^{N} \mathbf{w}_{ij}\mathbf{x}_i$ and the normalization denominator $\sum_{i=1}^{N} \mathbf{w}_{ij}$, can be easily dispatched to multiple GPUs and calculated separately in parallel. Thus, we first separate the input mesh into multiple GPUs for parallel computing and only communicate when calculating attention among eidetic states.

Suppose that the initial separation splits input mesh into #gpu GPUs and the integral representation $\mathbf{x} \in \mathbb{R}^{N \times C}$ is split into $\{\mathbf{x}^{(1)}, \cdots, \mathbf{x}^{(\#\mathrm{gpu})}\}$, where $\mathbf{x}^{(k)} \in \mathbb{R}^{N_k \times C}$ and $N_k$ denotes the number of mesh points dispatched to the $k$-th GPU. Correspondingly, the point-wise slice weights $\mathbf{w} \in \mathbb{R}^{N \times M}$ is also separated into #gpu components $\{\mathbf{w}^{(1)}, \cdots, \mathbf{w}^{(\#\mathrm{gpu})}\}$ with $\mathbf{w}^{(k)} \in \mathbb{R}^{N_k \times M}$. For clarity, we treat both multi-node-multi-GPU and single-node-multi-GPU configurations as a unified case. The calculation of eidetic states $\mathbf{s}_j$ in Eq. (1) can be equivalently rewritten into the following parallelism formalization:

$$\mathbf{s}_j = \frac{\sum_{i=1}^{N_1} \mathbf{w}_{ij}^{(1)}\mathbf{x}_i^{(1)} \oplus \cdots \oplus \sum_{i=1}^{N_{\#\mathrm{gpu}}} \mathbf{w}_{ij}^{(\#\mathrm{gpu})}\mathbf{x}_i^{(\#\mathrm{gpu})}}{\sum_{i=1}^{N_1} \mathbf{w}_{ij}^{(1)} \oplus \cdots \oplus \sum_{i=1}^{N_{\#\mathrm{gpu}}} \mathbf{w}_{ij}^{(\#\mathrm{gpu})}}$$

(5)

where $\oplus$ denotes the AllReduce operation (Patarasuk & Yuan, 2009), which aggregates the results from all processes. In practice, the $k$-th GPU first independently computes its partial sums for the numerator $(\sum_{i=1}^{N_k} \mathbf{w}_{ij}^{(k)}\mathbf{x}_i^{(k)})$ and denominator $(\sum_{i=1}^{N_k} \mathbf{w}_{ij}^{(k)})$ of $N_k$ points. Next, these partial results are synchronized across GPUs to compute the eidetic states $\mathbf{s}_j$, highlighted by blue-marked steps in Algorithm 1.

**Overhead analysis**  Consider the input $\mathbf{x} \in \mathbb{R}^{N \times C}$ with $N$ mesh points and $C$ channels. To handle large model

---

**Algorithm 1** Parallel Physics-Attention with Eidetic States

**Input:** Input features $\mathbf{x}^{(k)} \in \mathbb{R}^{N_k \times C}$ on the $k$-th GPU.
**Output:** Updated output features $\mathbf{x}'^{(k)} \in \mathbb{R}^{N_k \times C}$.
// drop $\mathbf{f}$ to save 50% memory.
Compute $\cancel{\mathbf{f}^{(k)}}, \mathbf{x}^{(k)} \leftarrow \mathrm{Project}(\mathbf{x}^{(k)})$
Compute $\tau^{(k)} \leftarrow \tau_0 + \mathrm{Ada\text{-}Temp}(\mathbf{x}^{(k)})$
Compute weights $\mathbf{w}^{(k)} \leftarrow \mathrm{Rep\text{-}Slice}(\mathbf{x}^{(k)}, \tau^{(k)})$
Compute weights norm $\mathbf{w}_{\mathrm{norm}}^{(k)} \leftarrow \sum_{i=1}^{N_k} \mathbf{w}_i^{(k)}$
Reduce slice norm $\mathbf{w}_{\mathrm{norm}} \leftarrow \mathrm{AllReduce}(\mathbf{w}_{\mathrm{norm}}^{(k)})$   $\mathcal{O}(M)$
Compute eidetic states $\mathbf{s}^{(k)} \leftarrow \frac{\mathbf{w}^{(k)\top}\mathbf{x}^{(k)} \cancel{\mathbf{f}^{(k)}}}{\mathbf{w}_{\mathrm{norm}}}$
Reduce eidetic states $\mathbf{s} \leftarrow \mathrm{AllReduce}(\mathbf{s}^{(k)})$   $\mathcal{O}(MC)$
Update eidetic states $\mathbf{s}' \leftarrow \mathrm{Attention}(\mathbf{s})$
Deslice back to $\mathbf{x}'^{(k)} \leftarrow \mathrm{Deslice}(\mathbf{s}', \mathbf{w}^{(k)})$
**Return** $\mathbf{x}'^{(k)}$

---

parameters in linear layers, tensor parallelism partitions the model parameters along the channel dimension. It reduces the memory consumption linearly but introduces an increase in communication overhead of $O(N)$. Attention-optimized methods like RingAttention leverage the FlashAttention concept to distribute the outer loop using a ring topology, which results in $O(N^2)$ communication complexity, while DeepSpeed-Ulysses, similar to tensor parallelism, partitions the data along feature dimensions and yields an $O(N)$ communication volume. However, all of the above methods are not feasible when handling million-scale meshes as the communication overhead shown in Figure 3(b) is unacceptable.

In parallel Transolver++, each GPU computes two partial sums of size $\mathcal{O}(MC)$ and $\mathcal{O}(M)$ separately, which are then synchronized across GPUs and cause a total communication volume of $\mathcal{O}(\#\mathrm{gpu} \times M(C+1))$, invariant to input size.

**Further speedup**  Also, we found that Transolver's code implementation is somewhat over-parameterized. Specifically, Transolver projects the data onto both $\mathbf{x}$ and $\mathbf{f}$, where $\mathbf{x}$ is used to generate slice weights, and $\mathbf{f}$ is combined with weights to generate physical states. This repetitive design

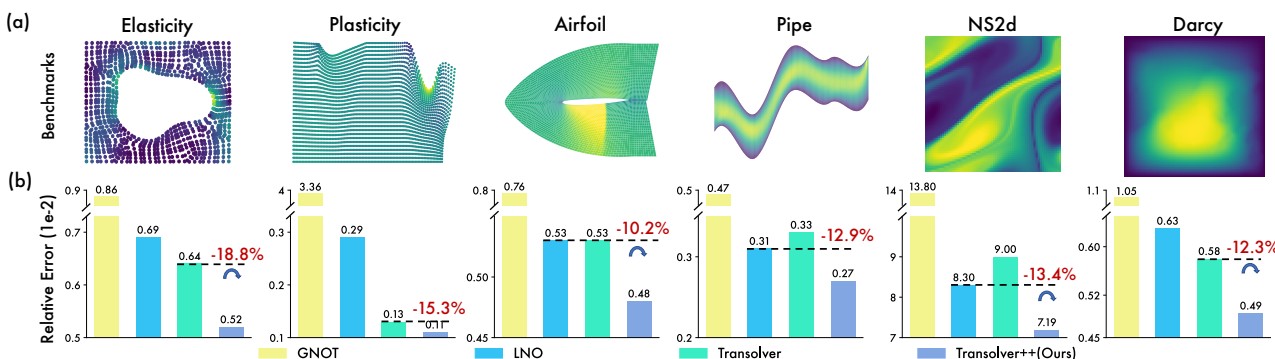

*Figure 4.* (a) Visualization of standard benchmarks covering a wide range of physics scenarios, from solid physics to fluid dynamics. (b) Relative errors on standard benchmarks of the top-4 models selected based on overall performance. Full results can be found in Table 8.

*Table 1.* Summary of experimental datasets, where #Mesh denotes the size of mesh points in each sample. Geo Type denotes the geometric structure of the mesh.

| TYPE | BENCHMARKS | GEO TYPE | #MESH |
|---|---|---|---|
| | NS2D | STRUCTURE | 4,096 |
| | PIPE | STRUCTURE | 16,641 |
| STANDARD | DARCY | STRUCTURE | 7,225 |
| BENCHMARKS | AIRFOIL | STRUCTURE | 11,271 |
| | PLASTICITY | STRUCTURE | 3,131 |
| | ELASTICITY | UNSTRUCTURE | 972 |
| INDUSTRIAL | AIRCRAFT | UNSTRUCTURE | ~300K |
| APPLICATIONS | DRIVAERNET++ | | ~2.5M |

brings double memory costs. In this paper, we find that eliminating **f** (marked gray in Algorithm 1) can simply and successfully increase the single-GPU input point capacity to 1.2 million, without sacrificing the model's performance.

# 5. Experiments

We extensively evaluate Transolver++ on six standard benchmarks and two industrial-level datasets with million-scale meshes, covering both physics and design-oriented metrics.

**Benchmarks** As summarized in Table 1, our experiments include both standard benchmarks and industrial simulations, which cover a broad range of mesh sizes. Specifically, the standard benchmarks include Elasticity, Plasticity, Airfoil, Pipe, NS2d, and Darcy, which are widely used in previous studies (Wu et al., 2024). To further evaluate the model's efficacy in real applications, we also perform experiments on industrial design tasks, where we utilized DrivAerNet++ (Elrefaie et al., 2024) for car design and a newly simulated AirCraft dataset for 3D aircraft design. In addition to the error of predicted physics fields, we also measure the model performance for design by calculating drag and lift coefficients from predicted physics fields.

**Baselines** We widely compare Transolver++ against more than 20 advanced baselines, covering various types of

approaches. These baselines include 12 neural operators, such as Galerkin (2021), LNO (2024), and GINO (2023a), some of which are specifically designed for irregular meshes; 4 Transformer-based PDE solvers, including OFormer (2023c), FactFormer (2023d), GNOT (2023), and Transolver (2024); and 4 graph-neural-network-based methods: GraphSAGE (2017), PointNet (2017), Graph U-Net (2019), and MeshGraphNet (2021). Transolver is the previous state-of-the-art model. During experiments, we found that some neural operators designed for grids perform poorly for large-scale irregular meshes. Therefore, we only report their performance on standard benchmark datasets.

## 5.1. Standard Benchmarks

**Setups** As shown in Figure 4(a), we compare the latest state-of-the-art neural PDE solvers with Transolver++ on six standard datasets covering a wide range of physics scenarios. For a fair comparison, we keep all model parameters within a fixed range. Specifically, in Transolver++, we set the model's depth as eight layers, and the feature dimension is 128 or 256, depending on the scale of the data. The number of slices is chosen from {32, 64} to trade-off between computational cost and model performance.

**Results** As presented in Figure 4(b), Transolver++ yields over 13% improvement w.r.t. the dataset-specific second-best baseline averaged from all six standard benchmarks, showing the efficacy of our proposed methods in handling complex geometries. To highlight the comparison, we only present top-3 baselines and Transolver++ in Figure 4, which are chosen based on the overall performance. Full results for other baselines are provided in Table 8 of the Appendix.

It is worth noticing that Transolver and Transolver++ significantly outperform other models in Elasticity, whose geometry is recorded as an unstructured point cloud. Going beyond Transolver, Transolver++ further boosts performance by learning eidetic physical states. As shown in Figure 2, Transolver++ can learn more diverse slice partitioning, thereby enabling more accurate physics learning. More analyses on

*Table 2.* Comparison on large geometries benchmarks. Relative L2 of the surrounding area (*Volume*) and surface (*Surf*) physics as well as drag and lift coefficient ($C_D$, $C_L$) is recorded, along with their coefficient of determination $R_D^2$ and $R_L^2$. The closer $R^2$ is to 1, the better.

| Model | DrivAerNet++ Full | | DrivAerNet++ Surf | | | AirCraft | | |
|---|---|---|---|---|---|---|---|---|
| | Volume ↓ | Surf ↓ | $C_D$ ↓ | $R_L^2$ ↑ | Surf ↓ | $C_L$ ↓ | $R_L^2$ ↑ | Surf ↓ |
| GraphSAGE (2017) | 0.328 | 0.284 | 0.282 | 0.859 | 0.294 | 0.040 | 0.988 | 0.109 |
| PointNet (2017) | 0.285 | 0.478 | 0.301 | 0.831 | 0.237 | 0.095 | 0.982 | 0.169 |
| Graph U-Net* (2019) | 0.241 | 0.260 | 0.272 | 0.876 | 0.193 | 0.063 | 0.953 | 0.161 |
| MeshGraphNet* (2021) | 0.529 | 0.422 | 0.260 | 0.870 | 0.209 | 0.038 | 0.993 | 0.113 |
| GNO* (2020a) | 0.510 | 0.664 | 0.252 | 0.882 | 0.196 | 0.031 | 0.991 | 0.129 |
| Galerkin* (2021) | 0.234 | 0.274 | 0.267 | 0.792 | 0.235 | 0.069 | 0.879 | 0.118 |
| Geo-FNO* (2022) | 0.718 | 0.892 | 0.288 | 0.831 | 0.291 | 0.243 | 0.903 | 0.395 |
| GINO (2023a) | 0.586 | 0.638 | 0.323 | 0.725 | 0.220 | 0.047 | 0.983 | 0.133 |
| GNOT* (2023) | 0.174 | 0.171 | 0.158 | 0.901 | 0.167 | 0.033 | 0.991 | 0.093 |
| LNO* (2024) | 0.180 | 0.203 | 0.208 | 0.855 | 0.195 | 0.091 | 0.992 | 0.137 |
| 3D-GeoCA* (2024) | 0.389 | 0.224 | 0.205 | 0.883 | 0.175 | 0.022 | 0.993 | 0.097 |
| Transolver* (2024) | 0.173 | 0.167 | 0.061 | 0.931 | 0.145 | 0.037 | 0.994 | 0.092 |
| **Transolver++ (Ours)** | **0.154** | **0.146** | **0.036** | **0.997** | **0.110** | **0.014** | **0.999** | **0.064** |
| Relative Promotion | 11.0% | 12.6% | 41.0% | - | 24.1% | 36.3% | - | 30.4% |

\* These models cannot directly handle million-scale meshes as the model input. Thus, to enable comparison, we split the input mesh of these models into several pieces, independently infer them, and concatenate separately inferred outputs as their final results.

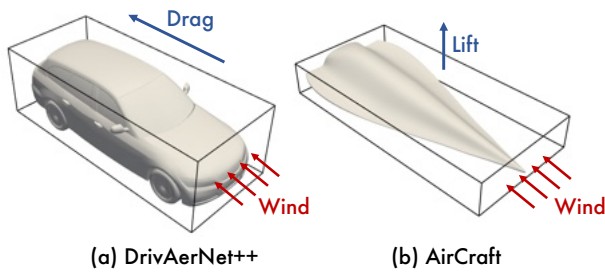

*Figure 5.* Car and aircraft design to predict drag and lift coefficient under extremely complex geometries with million-scale meshes.

learned physical states can also be found in Appendix C.

## 5.2. PDEs on Large Geometries

**Setups** To evaluate the performance in practical applications, we conduct experiments on two industrial datasets with million-scale meshes: DrivAerNet++ (Elrefaie et al., 2024) and Aircraft. The latter is newly presented by us, which is of high quality and simulated by aerodynamicists. To fully evaluate the model performance under different scales, we split DrivAerNet++ into two scales, one only with surface pressure (∼700k mesh points in each sample) and the other with full physics fields (2.5M mesh points).

In DrivAerNet++, the dataset is categorized into several distinct car types, including fastback, estateback, and notchback. To better demonstrate the generalizability of our proposed method across different geometries, we construct a representative subset by sampling from each category in proportion to its original distribution in the full dataset and

*Table 3.* Data distribution across different car types in the DrivAerNet++ dataset. Each category is sampled in proportion to its original distribution both in train set and test set. The other features, e.g. detailed or smooth wheel, are not under consideration.

| Type | DrivAerNet++ | | |
|---|---|---|---|
| | Fastback | Estateback | Notchback |
| # Train Cases | 54 | 72 | 54 |
| # Test Cases | 6 | 8 | 6 |
| Total | 60 | 80 | 60 |

ensures a fair and balanced evaluation. The detailed sampling statistics are presented in Table 3.

**Results** Table 2 demonstrates that Transolver++ achieved consistent state-of-the-art in all datasets with an average promotion of over 20%. In DrivAerNet++, our model is capable of handling 2.5 million meshes within 4 A100 GPUs and surpassing all other models by relative promotion of 11.0% and 12.6% on volume and surface field separately. In the surface field, our model still shows a prominent lead of 24.1% in DrivAerNet++ Surface and 30.4% in AirCraft.

We also observe that GNNs often degenerate quickly when handling large-scale meshes due to their geometric instability (Klabunde & Lemmerich, 2023). Additionally, Geo-FNO also performs poorly across nearly all datasets, as it attempts to map irregular meshes to uniform latent grids, a task that becomes especially difficult when the number of meshes scales up. These failures of our baselines further highlight the challenge of physics learning on million-scale geometries, while Transolver++ provides a practical solution, advancing an essential step to industrial applications.

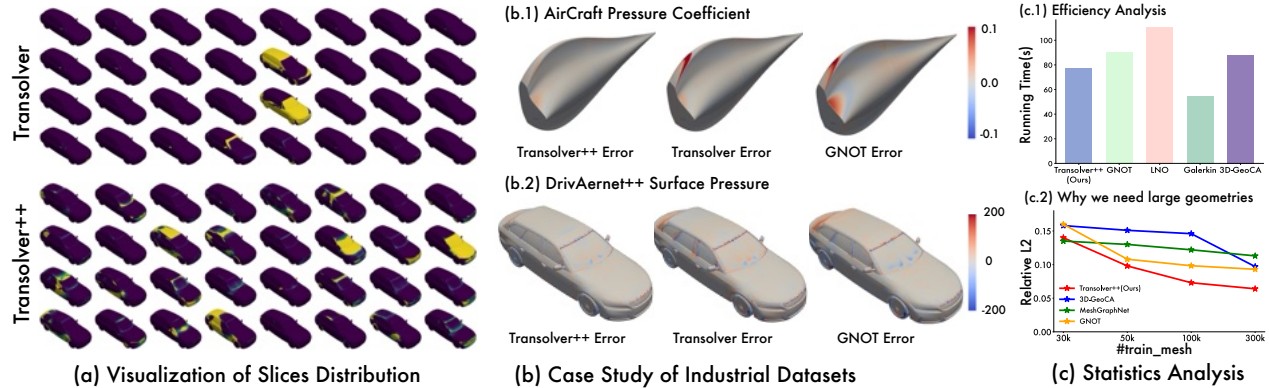

*Figure 6.* (a) Visualization of slice weight distributions. Transolver++ demonstrates a more diverse pattern than Transolver. (b) Error map of top-3 models on DrivAerNet++ and AirCraft. Transolver++ outperforms other models and captures some subtle variations. (c) Statistics analysis of efficiency in terms of running time and model performance under different scales of input geometries.

*Table 4.* Performance comparison of Transolver++ under patch and unpatch settings on the AirCraft dataset. $C_L$, $R_L^2$ and relative error on surface are all recorded. The best result for each metric is highlighted in **bold**, while the second-best is underlined.

| MODEL | AIRCRAFT | | |
|---|---|---|---|
| | $C_L \downarrow$ | $R_L^2 \uparrow$ | SURF$\downarrow$ |
| 3D-GEOCA | 0.022 | 0.993 | 0.097 |
| TRANSOLVER | 0.037 | 0.994 | 0.092 |
| TRANSOLVER++ (PATCH) | 0.017 | 0.997 | 0.072 |
| TRANSOLVER++ (UNPATCH) | **0.014** | **0.999** | **0.064** |

To ensure a fair and consistent comparison with existing methods, we additionally evaluate our model under the same patch-based input setting footnoted in Table 2, which is primarily adopted in other models to address memory constraints during training and inference phase. Specifically, we conduct experiments under both settings to assess the impact of spatial partitioning on model performance. As shown in Table 4, our model achieves better results in the unpatched setting, indicating that the parallelism framework effectively enhances overall performance with full data input. Notably, even when operating under the same patch setting, our model still outperforms other baselines, further validating the robustness and scalability of our approach across different computational settings and input scale.

As shown in Figure 6(b), our model has a lower relative error in most cases and excels in capturing the intrinsic physics variations in those regions with drastic changes, while other models tend to generate an over-smooth prediction, validating the effectiveness of learning eidetic states.

### 5.3. Model Analysis

In addition to model comparisons, we also conduct a series of analysis experiments to provide an in-depth understanding of our model and the necessity of large geometries.

*Table 5.* Ablations on AirCraft. Relative L2 and $R^2$ of lift coefficient are both recorded. *Ada-Temp* refers to adaptive mechanism in Eq. (3), *Reparameter* represents the gumbel-softmax operation with reparameterization trick for Eq. (4) and *Speedup* represents removing **f** in Algorithm for acceleration1.

| METHOD | AIRCRAFT | | |
|---|---|---|---|
| | $C_L \downarrow$ | $R_L^2 \uparrow$ | SURF$\downarrow$ |
| TRANSOLVER | 0.037 | 0.994 | 0.092 |
| + ADA-TEMP | 0.020 | 0.995 | 0.080 |
| + ADA-TEMP, SPEEDUP | 0.018 | 0.995 | 0.075 |
| + ADA-TEMP, REPARAMETER | 0.016 | 0.998 | 0.069 |
| **TRANSOVLER++ (OURS)** | **0.014** | **0.999** | **0.064** |

**Ablations** We conducted elaborative ablations on every component of Transolver++. As shown in Table 5, introducing the local adaptive mechanism significantly improves performance, which validates our motivation to learn eidetic states. With speed-up optimization and reparameterization techniques, the model achieves its best performance, fully demonstrating the effectiveness of our proposed design.

**Slice Analysis** As shown in Figure 6(a), Transolver++ can extract more diverse physical states than Transolver in million-scale meshes, enabling fine modeling for complex physics fields of driving cars. More visualization can be found in Appendix C. Moreover, in Table 6, we also calculate the KL divergence between learned slice weights and uniform distribution in different layers. These statistical results further demonstrate that Transolver++ can learn more diverse and varying slice distribution across all the layers.

**Efficiency analysis** In addition to GPU memory (Figure 1), we also measured the running time of different models on DrivAerNet++ datasets in Figure 6 (c.1). To ensure a fair comparison, all models' parameter sizes are well aligned in our experiments and we only compare Transolver++ with

*Table 6.* KL-divergence between learned slice weights and uniform distribution in different layers on Elasticity. We choose $\{1, 3, 5, 7\}$ layers and a higher value means a more diverse slice distribution generated by our method.

| MODEL | LAYER NUMBER | | | | AVG |
|---|---|---|---|---|---|
| | 1 | 3 | 5 | 7 | |
| TRANSOLVER | 0.056 | 5.599 | 3.474 | 0.383 | 3.885 |
| TRANSOLVER++ | **2.122** | **8.193** | **5.583** | **1.473** | **5.297** |

Transformer-based methods here, since GNNs struggle to handle million-scale meshes. We can find that Transolver++ strikes a favorable balance between performance and efficiency, outperforming most models in terms of speed. Moreover, at the same size of input mesh points, our method exhibits the lowest memory usage, highlighting its efficiency in handling large-scale data without sacrificing accuracy.

As shown in Table 7, to further highlight the advantages of deep learning-based approaches over traditional solvers, we compare our model with a traditional solvers, UnsCFD(McKenna et al., 1998), on the AirCraft dataset. Despite the time required for the entire training process, our model still proves to be more efficient when handling multiple simulation cases and consistently achieves better performance than traditional methods. Specifically, once trained, the model can generate high-fidelity predictions for new simulation cases within one second, bypassing the iterative convergence procedures inherent to traditional methods.

*Table 7.* Performance comparison of different models on the AirCraft dataset. $T_{\text{case}}$ denotes the time consumption per inference case, $T_{\text{train}}$ represents the total training time, and *Surf* indicates the relative surface prediction error. – means that this model does not involve a training process and has no associated training time.

| MODEL | AIRCRAFT | | |
|---|---|---|---|
| | $T_{\text{CASE}}$ ↓ | $T_{\text{TRAIN}}$ ↓ | SURF ↓ |
| UNSCFD | 5-6 HOURS | – | 0.173 |
| TRANSOLVER++ | < **1 SECOND** | 10 HOURS | **0.064** |

**Why we need large geometries** From the car mesh comparison in Figure 1(b), we can observe that large geometries differ significantly from smaller-scale ones in terms of details, which directly affects the accuracy of the simulation. While previous models (Li et al., 2021) claim to be resolution-invariant and aim to apply directly to large-scale geometries, our experiments in Figure 6 (c.2) show that without training on large meshes, the model will fall short in the fine-grained physics modeling, resulting in an insufficient and limited performance, which further necessitates the capability of handling larger geometries.

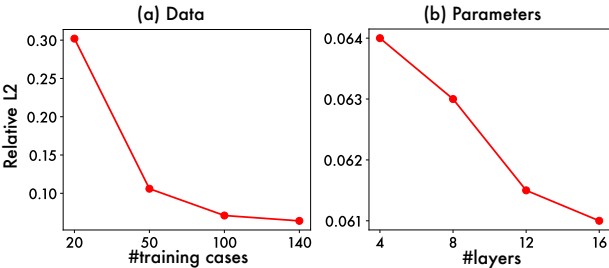

*Figure 7.* Evaluation of the model scalability in terms of data size and parameter size. Data size refers to the number of training cases, while parameter size is controlled by changing the number of network layers. Our default setting is 150 cases and 4 layers.

**Scalability** We evaluate the scalability of Transolver++ across different numbers of training samples and model parameters of different sizes by altering the number of layers. From Figure 7, we can find that our model can consistently benefit from large data and large models, revealing its potential to be the backbone of PDE-solving foundation models.

# 6. Conclusion

In pursuit of practical neural solvers, this paper presents Transolver++, which enables the first success in accurately solving PDEs discretized on million-scale geometries. Specifically, we upgrade the vanilla Transolver by introducing eidetic states and a highly optimized parallel framework, empowering Transolver++ with better physics learning and computation efficiency. As a result, our model achieves significant advancement in industrial design tasks, demonstrating favorable efficiency and scalability, which can serve as a neat backbone of PDE-solving foundation models.

# Impact Statement

This paper aims at advancing neural PDE solvers to industrial applications. Building on the previous advancement of Transolver, we present Transolver++ as an accurate neural solver for PDEs on million-scale geometries, which achieves consistent state-of-the-art in both standard benchmarks and industrial-level tasks with significant promotion. Also, Transolver++ pushes the single-GPU input size capacity to the million scale for the first time and achieves linear scalability with resources, showing favorable practicability to industrial manufacturing. Our work only focuses on the scientific problem, so we believe there is no potential ethical risk.

# Acknowledgements

This work was supported by the National Natural Science Foundation of China (U2342217 and 62021002), the BNRist Innovation Fund (BNR2024RC01010), and the National Engineering Research Center for Big Data Software.

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

# A. Full Results on Standard Benchmarks

Due to the space limitation of the main text, we present the full results of model performance on standard benchmarks here, as a supplement to Figure 4. As shown in Table 8, Transolver++ demonstrates superior performance across six standard PDE benchmarks, achieving the lowest relative L2 error in all PDE-solving tasks with an averaged relative promotion of 13%.

For models marked with an asterisk (*), we carefully reproduced the results by running the models more than three times and enabled fair and reliable comparison by maintaining the model parameter counts within a closely comparable range with other models. Specifically, we adjust LNO and reduce its default dimension from 256 to 192 in elasticity, as its parameter count is already three times larger than the other largest models. As discussed in previous studies (Wu et al., 2024), Transformer-based models usually benefit from large parameter sizes. This approach minimizes the potential influence of parameter variations and highlights the performance improvements achieved by our method under similar computation costs.

*Table 8.* Model performance on six standard PDE benchmarks is evaluated using the relative L2 error. The result marked with an asterisk (*) indicates a reproduced outcome, where the parameter counts and configurations of the baseline methods are carefully aligned to ensure a fair comparison. "/" means that the baseline cannot apply to this benchmark.

| MODEL | RELATIVE L2 | | | | | |
|---|---|---|---|---|---|---|
| | ELASTICITY | PLASTICITY | AIRFOIL | PIPE | NS2D | DARCY |
| FNO (2021) | / | / | / | / | 0.1556 | 0.0108 |
| U-FNO (2022) | 0.0239 | 0.0039 | 0.0269 | 0.0056 | 0.2231 | 0.0183 |
| GEO-FNO (2022) | 0.0229 | 0.0074 | 0.0138 | 0.0067 | 0.1556 | 0.0108 |
| U-NO (2023) | 0.0258 | 0.0034 | 0.0078 | 0.0100 | 0.1713 | 0.0113 |
| F-FNO (2023) | 0.0263 | 0.0047 | 0.0078 | 0.0070 | 0.2322 | 0.0077 |
| LSM (2023) | 0.0218 | 0.0025 | 0.0059 | 0.0050 | 0.1535 | 0.0065 |
| LNO* (2024) | 0.0069 | 0.0029 | 0.0053 | 0.0031 | 0.0830 | 0.0063 |
| GALERKIN (2021) | 0.0240 | 0.0120 | 0.0118 | 0.0098 | 0.1401 | 0.0084 |
| HT-NET (2022) | / | 0.0333 | 0.0065 | 0.0059 | 0.1847 | 0.0079 |
| OFORMER (2023C) | 0.0183 | 0.0017 | 0.0183 | 0.0168 | 0.1705 | 0.0124 |
| GNOT (2023) | 0.0086 | 0.0336 | 0.0076 | 0.0047 | 0.1380 | 0.0105 |
| FACTFORMER (2023D) | / | 0.0312 | 0.0071 | 0.0060 | 0.1214 | 0.0109 |
| ONO (2024) | 0.0118 | 0.0048 | 0.0061 | 0.0052 | 0.1195 | 0.0076 |
| TRANSOLVER (2024) | 0.0064 | 0.0013 | 0.0053 | 0.0033 | 0.0900 | 0.0058 |
| **TRANSOLVER++ (OURS)** | **0.0052** | **0.0011** | **0.0048** | **0.0027** | **0.0719** | **0.0049** |
| RELATIVE PROMOTION | 18.8% | 15.3% | 10.2% | 12.9% | 13.4% | 12.3% |

# B. Implementation Details

In this section, we provide detailed descriptions of the benchmarks, baseline methods, and implementation setups to ensure reproducibility and facilitate comparisons.

### B.1. Benchmarks

We evaluate our method on six standard PDE benchmarks, including Elasticity, Plasticity, Airfoil, Pipe, NS2D, and Darcy, as well as two industrial datasets, DrivAerNet++ and AirCraft. These benchmarks cover a wide range of physics simulation tasks, varying in complexity and geometry, and serve as a comprehensive benchmarks for assessing the effectiveness of neural PDE solvers as shown in Table 9. Here are the details of these datasets.

**Elasticity**   This benchmark is generated by the simulations of the stress field in a hyper-elastic solid body under tension, which is governed by a stress-strain relationship using the Rivlin-Saunders material model (Li et al., 2022). Each case involves a unit cell of 972 points with a void in the middle , clamped at the bottom edge and subjected to tension on the top. Elasticity contains a total of 1200 samples, with 1000 for training and 200 for testing.

**Plasticity**   This benchmark is generated by the simulation of a plastic forging problem, where a block is impacted by a frictionless die moving at a constant speed (Li et al., 2022). An elastoplastic constitutive model is adopted to model this physical system with 900 training and 80 testing samples. The input is the external force on every mesh with the shape of $101 \times 31$, and the output is the time-dependent deformation and mesh grid over 20 timesteps.

*Table 9.* Details of different benchmarks, including geometric type, number of mesh points, as well as the type of input and output, etc. The split of the dataset is also provided to ensure reproducibility, which is listed in the order of (training samples, test samples).

| TYPE | BENCHMARK | #DIM | #MESHES | #INPUT | #OUTPUT | SPLIT |
|---|---|---|---|---|---|---|
| STANDARD BENCHMARK | ELASTICITY | 2D | 972 | STRUCTURE | INNER STRESS | (1000, 200) |
| | PLASTICITY | 2D + TIME | 3131 | EXTERNAL FORCE | MESH DISPLACEMENT | (900, 80) |
| | AIRFOIL | 2D | 11271 | STRUCTURE | MACH NUMBER | (1000, 200) |
| | PIPE | 2D | 16641 | STRUCTURE | VELOCITY | (1000, 200) |
| | NAVIER-STOKES | 2D + TIME | 4096 | VELOCITY | VELOCITY | (1000, 200) |
| | DARCY | 2D | 7225 | POROUS MEDIUM | PRESSURE | (1000, 200) |
| INDUSTRIAL APPLICATIONS | DRIVAERNET++ | 3D | ∼700K | STRUCTURE | SURFACE PRESSURE | (190, 10) |
| | | | ∼2.5M | STRUCTURE | PRESSURE & VELOCITY | (190, 10) |
| | AIRCRAFT | 3D | ∼300K | STRUCTURE | 6 QUANTITIES | (140, 10) |

**Airfoil** This benchmark is generated from simulations of transonic flow over an airfoil, governed by Euler's equations (Li et al., 2022). The whole field is discretized to unstructured meshes in the shape of $221 \times 51$ as the input and the output is the corresponding Mach number on these meshes. The dataset includes 1000 training samples and 200 test samples that are based on the initial NACA-0012 shape.

**Pipe** This benchmark consists of simulations of incompressible flow in a pipe, governed by the Navier-Stokes equation with viscosity $\nu = 0.005$ (Li et al., 2022). The pipe has a length of 10 and a width of 1, with its centerline parameterized by cubic polynomials determined by five control nodes. The dataset also contains 1000 training and 200 testing samples, whose inputs are the mesh point locations $129 \times 129$ and outputs are the horizontal velocity field.

**Navier-Stokes** This benchmark consists of simulations of the 2D Navier-Stokes equations in vorticity form on the unit torus $(0, 1)^2$ (Li et al., 2021). The objective is, given the past velocity, to predict future velocity for 10 steps on discretized meshes in the shape of $64 \times 64$. The inputs are the velocity for the past 10 steps, while the outputs provide the future velocity for 10 timesteps. The dataset also includes 1000 samples for training and 200 for testing.

**Darcy** This benchmark consists of simulations of the steady-state Darcy Flow in two dimensions, governed by a second-order elliptic equation on the unit square (Li et al., 2022). The input is the structure of the porous medium and the output is the corresponding fluid pressure. The dataset contains 1000 training samples and 200 testing samples, generated using a second-order finite difference scheme on $421 \times 421$ uniform grids and later downsampled to $85 \times 85$.

**DrivAerNet++** This benchmark (Elrefaie et al., 2024) is a large and comprehensive dataset for aerodynamic car design, featuring high-fidelity computational fluid dynamics (CFD) simulations. It includes over 8,000 car designs with various configurations of wheel and underbody design. To ensure efficiency while maintaining diversity, we select 200 representative cases from these designs. Notably, in our experiments, DrivAerNet++ is then divided into two subsets with different levels of resolution. The first subset, as shown in Table 9, only consists of surface meshes, where each mesh point is characterized by its 3D position $(x, y, z)$, surface normal vector $(u_x, u_y, u_z)$, and signed distance function (SDF). The output for this subset is the surface pressure on these meshes. The second subset provides a full 3D pressure and velocity field, significantly increasing the dataset's complexity, with the number of mesh points reaching approximately 2.5 million and requiring the model to predict surface pressure, velocity and pressure of the surrounding area. The dataset consists of 190 training samples and 10 test samples, offering a rigorous benchmark for evaluating aerodynamic modeling at different levels of fidelity.

**AirCraft** This benchmark includes simulations of over 30 aircraft designs under 5 different incoming flow conditions, varying in Mach number, angle of attack, and sideslip angle. Unlike commonly used aerodynamics datasets such as Airfoil, the AirCraft dataset discretizes each aircraft into approximately 300,000 3D mesh points, offering a significantly higher resolution for capturing complex aerodynamic phenomena. Each mesh point is characterized by its spatial coordinates $(x, y, z)$ and surface normal vector, serving as the input. These high-fidelity computational fluid dynamics (CFD) simulations, conducted by aerodynamicists in an aircraft design institution, ensure precise and realistic aerodynamic modeling. The dataset requires predicting six key physical quantities: pressure coefficient $C_p$, fluid density $\rho$, velocity components ($u$, $v$, $w$), and pressure $p$. With 140 training cases and 10 test cases, this dataset presents intricate aerodynamic interactions, making it a rigorous benchmark for assessing model scalability and accuracy.

## B.2. Metrics

To comprehensively evaluate model performance across different datasets and prediction tasks, we adopt relative L2 error as the primary evaluation metric. For large-scale datasets, we evaluate both field and coefficient predictions separately. And we further introduce R-squared ($R^2$) score as an additional metric to assess the accuracy of coefficient predictions. In this section, we would explain how these metrics are applied to different dataset categories in detail.

### B.2.1. STANDARD BENCHMARKS

For standard benchmark datasets, model performance is assessed using the relative L2 error, which directly measures the difference between the predicted output and the ground truth. Given an output field $\hat{y}$ predicted by the model and the ground truth $y$, the relative L2 error is computed as:

$$\text{Relative L2} = \frac{\|\hat{\mathbf{y}} - \mathbf{y}\|_2}{\|\mathbf{y}\|_2}. \tag{6}$$

This metric on standard benchmarks is reported in the Table 8, providing a direct comparison across different models.

### B.2.2. LARGE-SCALE DATASETS

**Field and Coefficient Errors**  For large-scale datasets, we decompose the relative L2 error into two components to gain deeper insights into model performance. Since large-scale datasets often involve both surface and volume (refers to the surrounding area) data, we separately compute the relative L2 error for surface fields and volume fields. This distinction allows us to evaluate how well the model captures different types of physical information. In addition to predicting field values, some datasets require models to infer physical coefficients that characterize the feature of the system. The relative L2 error is also applied to these coefficients to measure prediction accuracy.

For example, in AirCraft, lift coefficient is calculated to measure the aerodynamic lift force on the body of an aircraft, which is a key metric to assess the lift performance of an aircraft under a certain flow condition. The lift coefficient is defined as:

$$C_L = \frac{1}{\frac{1}{2}\rho v_\infty^2 A} \int_S \left(-p\mathbf{n} \cdot \hat{\mathbf{l}} + \boldsymbol{\tau}\mathbf{n} \cdot \hat{\mathbf{l}}\right) \mathrm{d}S, \tag{7}$$

where $p$ is the pressure field on the surface, $\mathbf{n}$ is the unit normal vector of the surface, $\hat{\mathbf{l}}$ is the unit vector in the lift direction, $\boldsymbol{\tau}$ is the shear stress tensor and $S$ is the surface of the aircraft.

Also in car design, drag coefficient is a crucial metric to quantify the aerodynamic drag force on the body of a vehicle, which can be used to improve fuel efficiency and the performance of vehicles. The drag coefficient is defined as:

$$C_D = \frac{1}{\frac{1}{2}\rho v_\infty^2 A} \int_S \left(-p\mathbf{n} \cdot \hat{\mathbf{d}} + \boldsymbol{\tau}\mathbf{n} \cdot \hat{\mathbf{d}}\right) \mathrm{d}S, \tag{8}$$

where $p$ is the surface pressure, $\mathbf{n}$ is the unit normal vector of the surface, $\hat{\mathbf{d}}$ is the unit vector in the drag direction, $\boldsymbol{\tau}$ is the shear stress tensor and $S$ is the surface of the vehicle.

**R-squared Score for Coefficient Predictions**  To further evaluate the model's ability to predict physical coefficients, we introduce the R-squared ($R^2$) score as an additional metric. The $R^2$ score is computed as:

$$R^2 = 1 - \frac{\sum_i (y_i - \hat{y}_i)^2}{\sum_i (y_i - \bar{y})^2}, \tag{9}$$

where $\hat{y}_i$ is the $i$-th of the predicted coefficients, $y_i$ is the $i$-th ground truth, and $\bar{y}$ is the mean of the ground truth values. An $R^2$ score closer to 1 indicates better model performance, while lower values suggest less accurate predictions. This metric measures how well the model learns the physical quantitiy fields among all samples.

## B.3. Baselines and Implementations

We conduct extensive comparisons between Transolver++ and over 20 state-of-the-art baselines, encompassing a diverse range of methods for solving partial differential equations (PDEs). These baselines include typical neural operators,

*Table 10.* Implementation details of Transolver++ including training and model configuration. Training configurations are identical to previous methods (Wu et al., 2024; Hao et al., 2023; Deng et al., 2024; Elrefaie et al., 2024) and shared in all baselines. $\mathcal{L}_\mathrm{v}$ and $\mathcal{L}_\mathrm{s}$ refer to the loss on volume (surrounding area) and surface physics fields respectively.

| BENCHMARKS | TRAINING CONFIGURATION (SHARED IN ALL BASELINES) | | | | | MODEL CONFIGURATION | | | |
| --- | --- | --- | --- | --- | --- | --- | --- | --- | --- |
| | LOSS | EPOCHS | INITIAL LR | OPTIMIZER | BATCH SIZE | LAYERS $L$ | HEADS | CHANNELS $C$ | SLICES $M$ |
| ELASTICITY | | | | | 8 | | | 128 | 64 |
| PLASTICITY | | | | | 8 | | | 128 | 64 |
| AIRFOIL | RELATIVE L2 | 500 | $10^{-3}$ | ADAMW (2019) | 4 | 8 | 8 | 128 | 64 |
| PIPE | | | | | 4 | | | 128 | 64 |
| NAVIER–STOKES | | | | | 8 | | | 256 | 32 |
| DARCY | | | | | 4 | | | 128 | 64 |
| DRIVAERNET++ FULL | $\mathcal{L}_\mathrm{v} + \mathcal{L}_\mathrm{s}$ | | | ADAM | | | | | |
| DRIVAERNET++ SURF | $\mathcal{L}_\mathrm{s}$ | 200 | $10^{-3}$ | (2015) | 1 | 4 | 8 | 256 | 32 |
| AIRCRAFT | $\mathcal{L}_\mathrm{v} + \mathcal{L}_\mathrm{s}$ | | | | | | | | |

Transformer-based PDE solvers, and graph neural networks (GNNs) and are tested under the same training configurations as shown in Table 10. To ensure a rigorous comparison, we obtain the open-source implementations of these models and carefully verify their consistency with the original papers before training.

### B.3.1. STANDARD BENCHMARKS

We try to be as loyal to the original settings of baselines as possible. However, when the model parameters count is too large to conduct a fair comparison, we would change either the number of blocks or the hidden dimension to ensure a fair comparison. As we mentioned before, Transformer-based models usually benefit from large parameter sizes (Wu et al., 2024). Thus, the alignment of parameter size is essential to control the variable and highlight the performance difference caused by architecture design. Here are detailed implementations.

Specifically, we adjust LNO and reduce its default dimension from 256 to 192 in elasticity, as its parameter count is already three times larger than the other largest models, while other models' settings remain unchanged.

Besides, we carefully review the original papers of these baseline models to make sure that they are fully tuned on their hyper-parameters. For all the models whose data settings align exactly with ours, such as FNO (Li et al., 2021), Geo-FNO (Li et al., 2022), GNOT (Hao et al., 2023), OFormer (Li et al., 2023c), Transolver (Wu et al., 2024), we directly adopt the results reported in their respective publications. For other models, we refer to the results presented in LSM (Wu et al., 2023) and Transolver (Wu et al., 2024), as these works have conducted comprehensive and rigorous hyper-parameter tuning to ensure a fair and reliable comparison.

### B.3.2. INDUSTRIAL APPLICATIONS

For DrivAerNet++ and AirCraft datasets, most of the baseline models are unable to handle million-scale meshes directly. To address this, we subsample the dataset to 50k meshes point and reconstruct the meshes using K-Nearest Neighbors (KNN) (Peterson, 2009) method to preserve essential geometric relationships.

Specifically during training, we apply a random subsampling strategy at the beginning of each epoch and reconstruct meshes for only once in each case to maintain consistency of the total step count compared to other baselines capable of handling million-scale meshes. For testing, we perform a complete subsampling operation, obtaining multiple partial predictions across different subsampled sets. These predictions are then aggregated and concatenated to reconstruct the full output for each test case. The relative L2 error is subsequently computed on the concatenated predictions, and the final results are reported in Table 2. Furthermore, lift and drag coefficient are also computed based on the reconstructed outputs following their respective formulations in Eq. (8) and Eq. (9).

In terms of model configurations, each model has been extensively tuned on their hyperparameters, especially graph neural networks (GNNs) for their extreme instability (Klabunde & Lemmerich, 2023) when dealing with million-scale meshes. To minimize human bias, we employ a standardized tuning strategy by systematically changing the hidden dimension from {64, 128, 256, 512}, the number of layers from {2, 4, 6, 8, 10} along with tuning model-specific hyperparameters. During this process, we observe that almost all GNNs experienced geometric instability (Morris et al., 2023) when applied to the

DrivAerNet++ Full dataset with approximately 2.5 million mesh points per case. Out of all baselines, Transsolver achieves the second-best performance, benefiting from its Slice-Deslice mechanism, which improves its stability and scalability, while Transolver++ surpasses it by introducing eidetic physical states and a parallelism framework that can directly handle million-scale meshes with enhanced efficiency. In Transolver++, we set the number of slices to 32 and its channels to 256 with 4 layers of Transolver++ block to balance efficiency and performance. And we only need to adopt parallelism to Transolver++ for the DrivAerNet++ Full benchamrk on only 4 A100 GPUs at most.

## C. More Visualizations

As a supplement to Figure 6 of the main text, in this section, we will provide detailed visualization of the eidetic physical states of Transolver++ as well as case study showcases on different datasets.

### C.1. Eidetic States Visualization

Here, for simplicity and clarity, we present visualizations of the eidetic physical states on four representative benchmarks: DrivAerNet++ Surf, Airplane, Airfoil, and Elasticity. We further compare these states with those learned by Transolver, demonstrating the effectiveness of our approach in capturing eidetic states under complex physical geometries. All visualizations are extracted from the final layer of each model to provide a direct comparison of their learned representations.

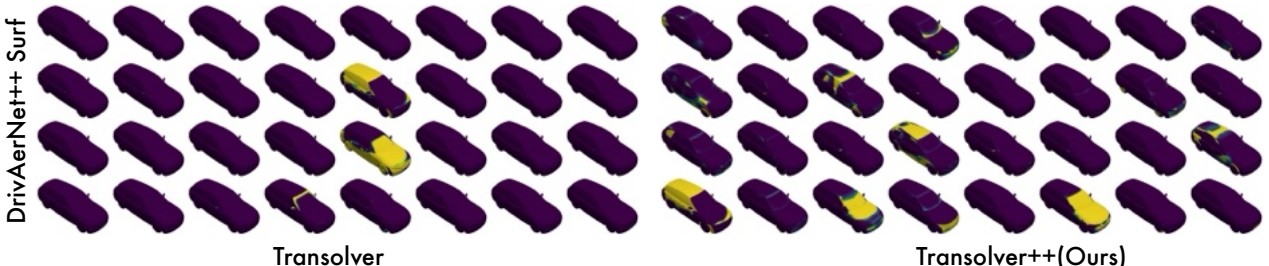

*Figure 8.* Visualizations of 32 physical or eidetic states learned in the final layer of models on DrivAetNet++ Surface. Transolver and Transolver++ are both plotted for a clear comparison. The lighter color means a higher weight in the corresponding physical state.

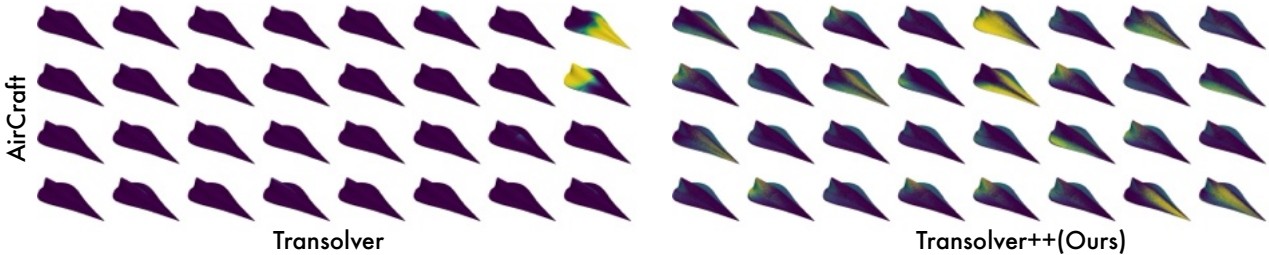

*Figure 9.* Visualizations of 32 physical or eidetic states learned in the final layer of models on AirCraft. Transolver and Transolver++ are both plotted for a clear comparison. The lighter color means a higher weight in the corresponding physical state.

### C.2. Case Studies

In addition to the Figure 6 in the main text, here we provide more showcase comparisons in Figure 14, 15 and 16. Specifically, we compare our model with the strong baselines in the respective datasets, where Transolver is selected for the comparison in standard benchmarks and both Transolver and GNOT are compared in industrial datasets.

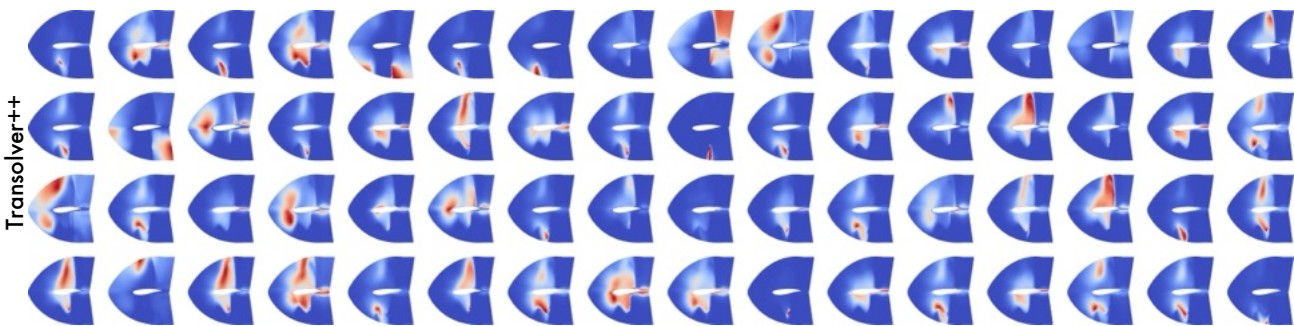

*Figure 10.* Visualizations of 64 learned eidetic states of the final layer in Transolver++ on Airfoil.

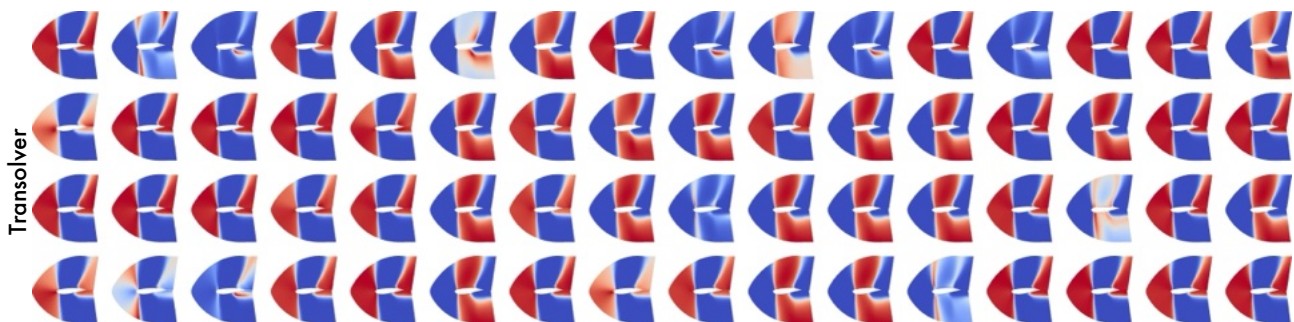

*Figure 11.* Visualizations of 64 learned physical states of the final layer in Transolver on Airfoil.

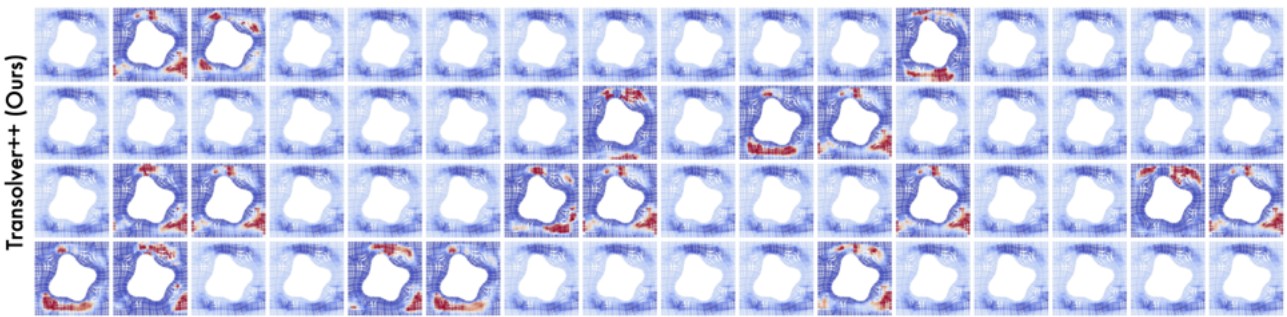

*Figure 12.* Visualizations of 64 learned eidetic states of the final layer in Transolver++ on Elasticity.

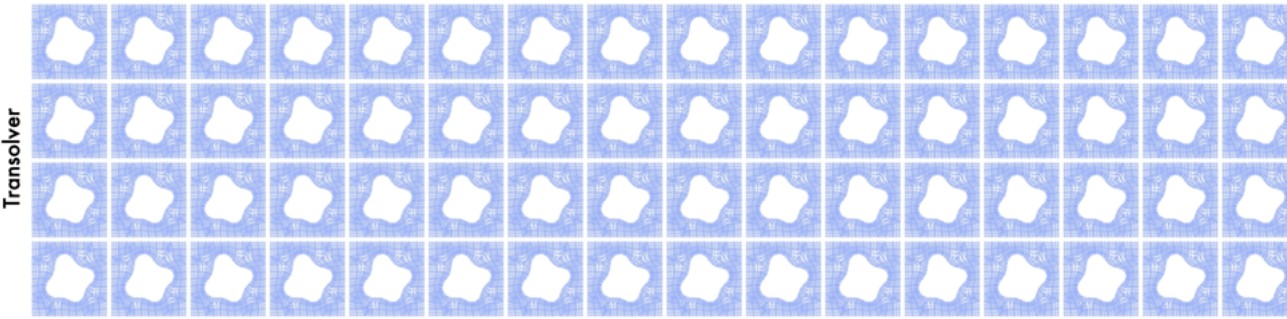

*Figure 13.* Visualizations of 64 learned states of the final layer in Transolver on Elasticity.

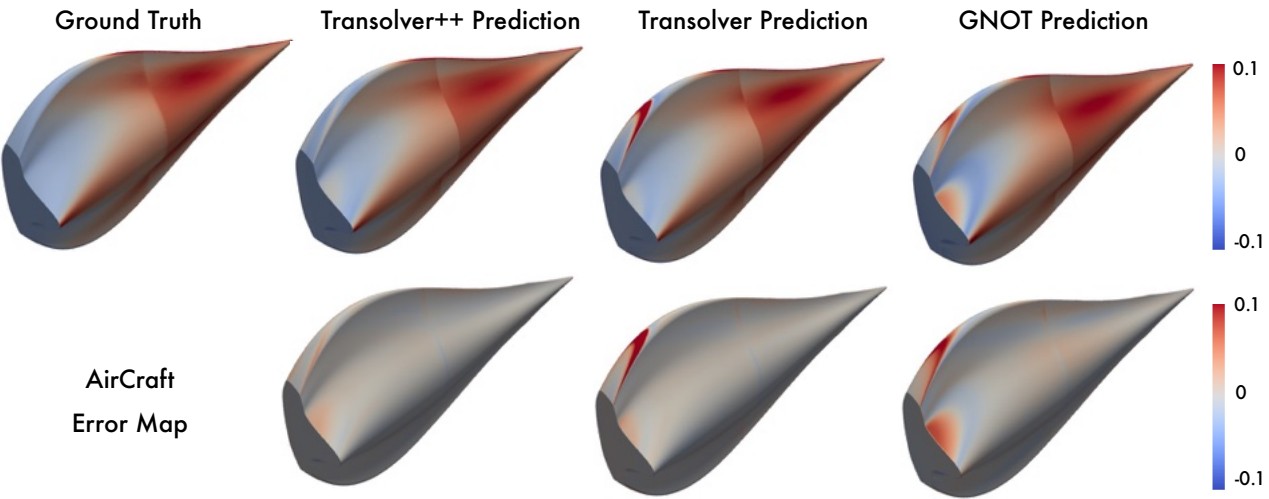

*Figure 14.* Showcase comparison with Transsolver and GNOT on AirCraft. A lighter color in the error map indicates a better performance.

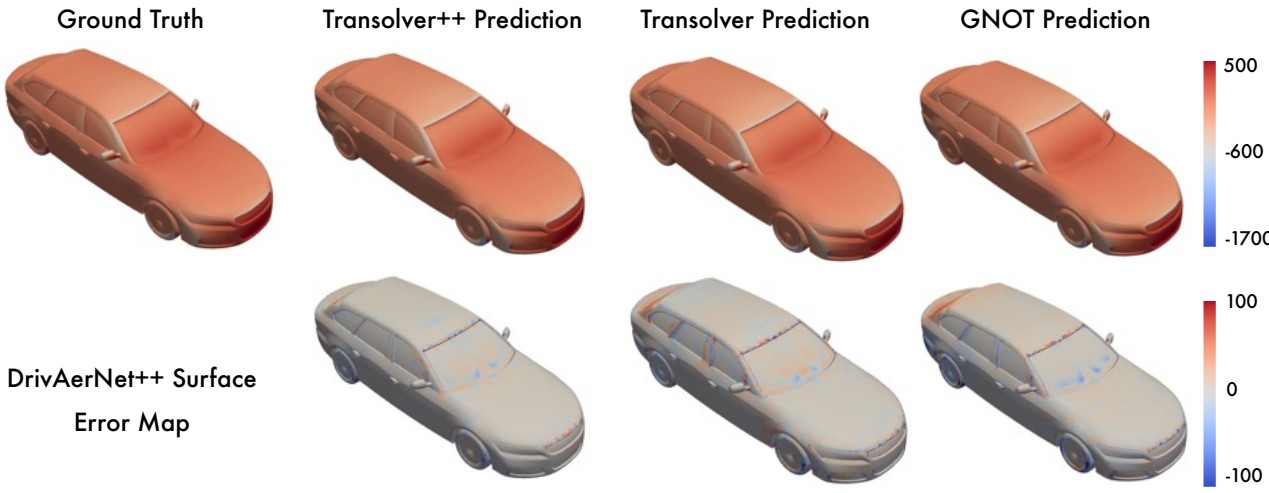

*Figure 15.* Showcase comparison with Transsolver and GNOT on DrivAerNet++ Surface. A lighter error map means better performance.

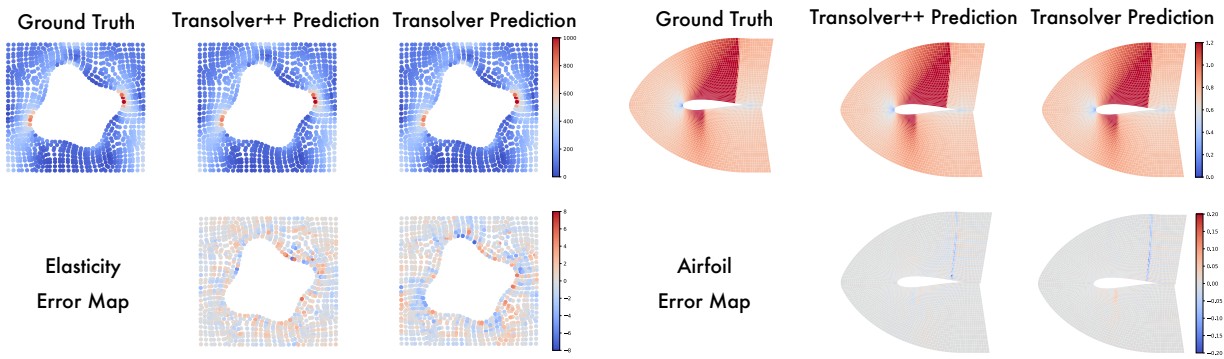

*Figure 16.* Showcase comparison with Transsolver on Elasticity and Airfoil. A lighter color in the error map indicates a better performance.

