# OpenReview forum: "Transolver++: An Accurate Neural Solver for PDEs on Million-Scale Geometries"
_ICML.cc/2025/Conference — ICML 2025 poster_

### Official Review · Reviewer_oAjG · 2025-02-21

**Overall Recommendation:** 2

**Summary:**

This paper presents two extensions to the Transolver architecture, for learning PDEs in systems with high resolution data. First, it introduces changes in how the weights for the "physical states" are computed, to achieve more peaky distributions. And second, it presents a Multi-GPU implementation of Transolver which reduces the communication overhead.

**Claims And Evidence:**

Yes. The main claims are extending Transolver to work with larger mesh sizes, and multi-GPU optimizations. The paper shows evidence for both.

**Essential References Not Discussed:**

References are fine.

**Experimental Designs Or Analyses:**

The most important experiments are there, all claims and novel method components have matching experiments.

One issue I see it that the analysis on the new datasets is a bit misleading-- here, baselines are inferred (and maybe trained?) on patches and stitched together, while Transolver++ is not. The paper does mention this, but a) it's easy to miss on a quick glance and b) since the impact to MSE is very much unclear, it's hard to know what to even take from this comparison.

And, as stated above, the paper could use clearer analysis to what exactly the issue with weight assignment is, and how or how not Transolver++ solves this. The hypothesis provided doesn't seem to be backed by the visualizations shown, and the only evidence is improved downstream performance.

**Methods And Evaluation Criteria:**

The paper largely reuses the baseline methods and datasets from the Transolver paper, and adds two further datasets. While not adding that much new over the previous paper, this is a broad coverage of different PDE systems and many relevant transformer and GNN based competing architectures which is nice to see.

I'm less sure about the method; the changes to weight computation seem to be a bit unprincipled.
It's not really clear to me what the goal exactly is-- from the text it sounds like the Transolver weights are too uniform for large meshes, and a more peaky distribution is desired (ideally assigning each mesh point to exactly one "physical state"). The obvious solution for this would be an annealing schedule for the temperature during training.
Instead, there's a complicated two-stage solution, a learned temperature per input point, and then differentiable sampling. The text doesn't quite explain why this makes sense.
1. Why would we want to spread some input features over several physical states, but not others?
2. Why would we want to use sampling for this? There's only one set of weights per input point.
3. Is the sampling seed fixed (i.e. each training step gets the same element given an unchanged weight distribution) or not?
4. What about stability? Both learning temperature and differentiable sampling seem like they would make training much more instable and seed-dependent.
And finally, from a look at the weight visualization in paper and appendix is seems like the problem might be something else entirely-- not the uniform distribution of a given input state, but the distribution over output _physical states_. Transolver doesn't seems to use all the physical states available, and instead assigns the entire input to 2 states. Transolver++ seems marginally better at this, but there isn't anything in the proposed algorithm which directly addresses the issue. This is a much harder (but also much more interesting) problem, if this indeed turns out to be the issue there's plenty of literature on differentiable class assignment to look at.

**Other Comments Or Suggestions:**

The paper uses quite a few confusing/vague terms and phrases, e.g. "massive mesh points", "overwhelm the learning process", "completely model the assignment process from points to certain physical states". In particular, "eidetic states" is an odd choice, I had never heard of this word before, and the dictionary definition doesn't match what the paper likely wants to convey.
So if the paper is accepted, I would advise an editing pass with this in mind.

# After rebuttal
The rebuttal did add a few new datapoints and explanations, which is appreciated. But the main issues I pointed out in my review still stand, and based on the rebuttal it didn't sound like the authors will address these in an updated version of the paper. So I think it's probably better if the paper went for a major revision & resubmit. Hence I will keep my score.

**Other Strengths And Weaknesses:**

Strengths: The paper does show empirical performance improvement on a wide set of benchmarks, including very large domains.
Weaknesses: The contributions are quite minor, and I'm not sure if we can learn much from the paper which will transfer to other work. As stated above, the weight computation is rather unprincipled and it's unclear why and how it works. The multi-GPU optimization is nice to see, but a bit of an implementation detail, and highly specific to this exact method.

**Questions For Authors:**

See "methods and evaluation".

**Relation To Broader Scientific Literature:**

The paper is an extension to the Transolver paper. It shows some performance improvement over Transolver, particularly for high-res problems. However, both contributions (improved weight computation, and multi-GPU optimization) are relatively minor, and quite niche-- the only really apply to the Transolver architecture and would not transfer to any other model for learned PDEs.

**Theoretical Claims:**

No theoretical claims/proofs in this paper.

---

> ### Author Rebuttal · Authors · 2025-04-01
>
> Many thanks to Reviewer oAjG for your instructive reviews.
>
> > **Q1:** It's not clear what the goal exactly is. The problem might be something else entirely: the distribution over output physical states.
>
> **(1) Our goal is to local-adaptively control each point’s state distribution.**
>
> As stated in lines 210-218, neither overly peaky nor smooth weights are desirable. As shown in Fig.3(c) and stated in its caption, points in slow varying regions should have sharp distribution, while those in fast varying regions with higher uncertainty expect a relatively smooth distribution.
>
> **(2) We respectfully disagree with “the problem might be something else."**
>
> We appreciate the reviewer’s careful observation. However, as shown in Figure 13 of Appendix (overly uniform assignments) and Figure 8 left of Appendix (Collapsed into two states), both cases illustrate unreasonable point-wise state assignments, failing to meet our goal above.
>
> These cases reveal an underlying issue: **the lack of adaptive point-wise assignment**, rather than the "distribution over output physical states." We believe our work directly addresses this core limitation of Transolver.
>
> **(3) New experiments on annealing.**
>
> As per your request, we conduct ablation studies (see Q2(1) of Reviewer dnan) with annealing strategies. This further verifies the statements above.
>
> > **Q2:** Why spread some input features over several states, not others?
>
> Again, our goal is point-wise adaptive distributions, not just peaky ones. A more uniform distribution is more suitable, especially for fast-changing regions with higher uncertainty.
>
> > **Q3:** Why use sampling? Is the sampling seed fixed? Stability
>
> As shown in Eq.(5) and Algo 1, we don't perform explicit sampling during training or inference. Instead, we generate new state weights using Gumbel-softmax, which enables the model to explore more diverse state assignment distribution. With a fixed seed at the beginning, each training step yields different distributions for the same element. This will not affect the training stability. Please see: https://anonymous.4open.science/r/ICML_rebuttal_3035-FD1C
>
> > **Q4:** The contributions are quite minor, cannot transfer. Multi-GPU optimization is nice, but a bit of an implementation detail.
>
> As you mentioned, Transolver++ is specific to Transolver, which is why we name it "Transolver++," not others. However, the considered questions and design principle of our paper are valuable to this community. Here is the summary.
>
> |Our design|Transferable Insights|
> |-|-|
> |local adaptive mechanism|PDE needs local-adaptive representations|
> |parallelism framework|Parallelism is an effective way to large-scale tasks|
> |Industrial-level experiment|Encourage the community to tackle large geometries|
>
> **(1) Q1 can demonstrate insights behind local adaptive mechanisms.**
>
> **(2) Parallelism framework stems from in-depth investigation.**
>
> The "multi-GPU optimization" design stems from our in-depth insights into Transolver and PDE-solving tasks, not only "implementation detail," where we carefully investigated which representation should be parallel computed or communicated. Without these efforts, Transolver++ cannot achieve the lowest communication overhead and presents linear scalability now.
>
> It is also the first efficient parallelism framework in neural operators for PDE solving, which can serve as a starting point for exploring model-system co-design in solving PDEs. Thus, the contribution of the parallelism framework should not be underestimated.
>
> **(3) Outstanding performance on industrial tasks can advance research to real applications. A new dataset of AirCraft will be made public.**
>
> > **Q5:** New datasets analysis is a bit misleading, baselines are on patches and stitched.
>
> Thanks for your detailed review. We will highlight our training setting in the revision.
>
> **(1) We have to conduct patch-wise training, otherwise all prior methods are inapplicable.**
>
> As noted in Table 2's footnote, prior methods cannot handle million-scale data and require patch-wise training. Since this is the only way (we can figure out) to measure other models on DrivAerNet++, we believe this comparison exactly shows their capability in large-scale tasks.
>
> This unaligned setting highlights the advantage of Transolver++ in large-scale tasks (parallelism framework is our key contribution), instead of being considered meaningless.
>
> **(2) Comparison under the same setting**
>
> We also provide results of aligned settings in six standard benchmarks and AirCraft. We will further clarify this in the revision. We also trained Transolver++ under the patch setting, which is still the best compared to other models.
>
> |DrivAerNet++|$C_D$|$R_L^2$|Surf|
> |-|-|-|-|
> |Transolver++ (patch)|0.017|0.997|0.072|
> |Transolver++ (unpatch)|0.014|0.999|0.064|
>
> > **Q6:** Confusing/vague terms and phrases.
>
> Many thanks. "Mesh points" are common in PDE-solving and refer to vertices. We will carefully revise other terms in the revision.

---

### Official Review · Reviewer_VjMs · 2025-03-03

**Overall Recommendation:** 4

**Summary:**

This paper introduces Transolver++, an extension of Transolver, designed to handle million-scale geometries in PDE solution operator learning. Building on the physics attention mechanism proposed in Transolver, which learns underlying physical states, this work presents two key advancements-a local adaptive mechanism and a parallelized implementation. The local adaptive mechanism enables Transolver++ to learn a per-point temperature used in the softmax function to determine the probability of a point belonging to a particular (learned) physical state. The parallelized implementation leverages Transolver’s design by synchronizing only the normalizer and physical states computed from a batch of points assigned to a single GPU, minimizing inter-GPU communication overhead. Transolver++ achieves a 13% performance improvement on standard PDE benchmarks compared to previous methods and a 20% performance gain on more challenging benchmarks with large-scale geometries, which are 100 times larger than those used in prior studies.

## Update After Rebuttal

I believe the concerns raised in my initial review are adequately addressed in the author rebuttal. While I acknowledge the points made in other reviews regarding the limited technical novelty, I find that this work offers a valuable perspective on scaling neural operators to handle PDEs on larger and more complex domains—a step toward real-world applicability. Therefore, I will maintain my original rating.

**Claims And Evidence:**

Yes, this paper identifies two major limitations of its predecessor, Transolver: the degeneration of learned physical states when applied to large-scale geometries and the lack of multi-GPU support. These issues are addressed in the main text with a detailed exposition. The authors' claims are well-supported by experimental results, which demonstrate clear performance improvements on standard PDE benchmarks as well as more challenging benchmarks, including the recently proposed DrivAerNet++ dataset, which involve complex irregular geometries.

**Essential References Not Discussed:**

As noted in the “Experimental Designs or Analyses” section, I would like to see the authors' discussion on UPT [1] and its follow-up [2], which extends the method to industry-level simulations. While I do not expect an empirical comparison, these works appear highly relevant to the scope of this study. Including them in the discussion would improve the completeness of the paper.

**References**

1. Universal Physics Transformers: A Framework For Efficiently Scaling Neural Operators, Alkin *et al.*, NeurIPS 2024
2. NeuralDEM-Real-time Simulation of Industrial Particulate Flows, Alkin *et al.*, arXiv 2024

**Experimental Designs Or Analyses:**

This paper validates the proposed method using commonly used PDE benchmarks, advancing the state of the art. The use of PDE dataset containing large-scale geometries is appropriate given the core objective of this work—developing a neural operator capable of handling such geometries. The authors include a wide range of baselines in qualitative and quantitative comparisons, covering both graph-based and transformer-based methods. Most of the experimental details are explained in the appendix.

While the text is well-written overall, there are some points that remain unclear to me:

1. How generalizable is the proposed method across different geometries, particularly in experiments involving car designs from the DrivAerNet++ dataset? While the paper mentions that 200 representative designs were selected, it would be beneficial to provide a statistical analysis of these shapes, including the distribution of car types (e.g., estateback, fastback, or notchback), to better assess the method's generalization capability.
2. Regarding the baselines, I am curious how this method compares to Universal Physics Transformers (UPT) [1], another recent transformer-based neural operator that claims to handle large-scale geometries. UPT aggregates local physics fields at a small number of supernodes and applies an attention mechanism among them, which, in my view, is conceptually similar to Transolver and its extension in this work. However, UPT is not mentioned in either the related work or experiment sections. I would like to hear the authors' perspective (*e.g.*, how these methods are different) on this method.

**Methods And Evaluation Criteria:**

Yes, the proposed method aligns with the neural operator literature and is validated using well-known benchmarks, employing the relative $L_2$ error metric, which is standard in the field.

**Other Comments Or Suggestions:**

In line 300, the text states “previous studies,” but only a single work is cited.

**Other Strengths And Weaknesses:**

I have no further comments on the strengths and weaknesses.

**Questions For Authors:**

I would like to asks the authors the following questions:
1. Could you provide more details on the statistical distribution of design parameters for car shapes in the training and test splits? A summary of how the shapes differ between these splits would help evaluate the generalizability of the proposed method.
2. How does this Transolver-based method differentiate itself from the Universal Physics Transformers (UPT) [1] approach? Have the authors experimented with UPT, or do they have a high-level understanding of why one method might outperform the other?
3. Based on my understanding, Transolver++ introduces a certain degree of stochasticity when mapping a point to multiple physical states, as described in lines 190–193 and Equation (4). Is this sampling performed only during training, or does it also occur during inference?
4. In line 267, what does the variable $\mathbf{f}$ represent? I could not find any mention of its meaning or usage in the Transolver paper either.

**References**
1. Universal Physics Transformers: A Framework For Efficiently Scaling Neural Operators, Alkin *et al*., NeurIPS 2024

**Relation To Broader Scientific Literature:**

The idea proposed in this work could advance deep learning for large-scale scientific computing, which often involves simulating complex systems discretized at high spatial and temporal resolutions.

**Theoretical Claims:**

This paper does not present any theoretical claims requiring review.

---

> ### Author Rebuttal · Authors · 2025-04-01
>
> Many thanks to Reviewer VjMs for your detailed and instructive review.
>
> > **Q1:** How generalizable is the proposed method across different geometries, particularly in experiments involving car designs from the DrivAerNet++ dataset? Could you provide more details on the statistical distribution of design parameters for car shapes in the training and test splits?
>
> Thank you for your constructive question, which helps to improve our work and analysis. We follow the sampling ratios provided in Figure 8 of the original DrivAerNet++ paper, using equal numbers of samples for both WW and WWC (wheels open detailed/closed):
>
> |Type|Fastback|Estateback|Notchback|
> |-|-|-|-|
> |#train cases|54|72|54|
> |#test cases|6|8|6|
> |Total|60|80|60|
>
> **Training and testing share the same distribution**, with no car-type shift between splits.
>
> Transolver++ shows **consistently strong performance across all categories**, confirming its robustness and generalization. A summary will be added in the revised version.
>
> |Type|Fastback|Estateback|Notchback|
> |-|-|-|-|
> |GNOT|0.167|0.143|0.190|
> |Transolver++|**0.112**|**0.108**|**0.109**|
>
> > **Q2:** UPT is a conceptually similar method and is not mentioned in the related work. How does this Transolver-based method differentiate itself from UPT and its follow-up NeuralDEM? Have the authors experimented with UPT, or do they have a high-level understanding of why one method might outperform the other?
>
> Thank you for your valuable suggestion. It prompted us to reflect on the distinction between Transolver-based methods and UPT, representing important advances in neural solvers for scientific problems.
>
> **(1) Conceptual Difference**
>
> First, we note that Transolver (ICML 2024) predates UPT (NeurIPS 2024).
>
> Second, the modeling paradigms differ: UPT decouples the encoder and decoder, enabling flexible querying across arbitrary scales, while Transolver-based models are fully end-to-end, enabling direct supervision on physical states and achieving stronger performance on fixed meshes (e.g., AirCraft).
>
> Third, **UPT compresses geometry via supernodes** into a global latent without explicit state assignments. **NeuralDEM** builds on this with primary quantity modeling and scalar control. In contrast, **Transolver++ learns soft point-to-state assignments**, enabling **interpretable adaptive assignments** based on shared physical behavior.
>
> Fourth, UPT evolves dynamics entirely in the global latent space. Transolver-based models apply **slice-deslice at every layer**, supporting **progressive refinement of physical states** and stronger geometry-physics coupling, with enhanced interpretability and control.
>
> **(2) Empirical Comparison**
>
> As our focus was on end-to-end architectures, UPT was not included in our original experiments. During the rebuttal period, we trained UPT on the AirCraft dataset.
>
> We observed that UPT has ~**10× more parameters** than Transolver++ (14.3M vs. 1.7M), which may lead to **slower convergence**. Additionally, its lack of explicit physical modeling for nodes may limit its ability to capture fine-grained structures. As a result, UPT underperforms compared to Transolver++ in this setting.
>
> |Model|Cp↓|$R_L^2$↑|Surf↓|
> |-|-|-|-|
> |UPT (Alkin et al., NeurIPS 2024)|0.035|0.994|0.112|
> |GraphViT (Janny et al., ICLR 2023)|0.041|0.990|0.130|
> |Point Transformer v3 (Wu et al., CVPR 2024)|0.045|0.987|0.145|
> |Transolver++|**0.014**|**0.999**|**0.064**|
>
> We sincerely apologize for not including UPT in the related work section earlier. A detailed comparison and proper citation will be added in the revised version.
>
> > **Q3:** Is this sampling performed only during training, or does it also occur during inference?
>
> As shown in Eq.(5) and Algo 1, stochasticity occurs in both training and inference, which enables the model to explore a more diverse state assignment distribution. Also, we provide train log plots at https://anonymous.4open.science/r/ICML_rebuttal_3035-FD1C to show the stability of Transolver++.
>
> > **Q4:** What does the variable f represent?
>
> As we stated in Lines 267-270 right column of main text, we have stated that "Also, we found that ..., where x is used to generate slice weights, and f is combined with weights to generate physical states." In all, f represents geometric features while x, as other features, is used to generate weights in the github implementation of Transolver. We will make this clearer in the revised paper. Notably, this design significantly reduces computational complexity and enables the first practical application of deep learning to large-scale industrial simulations.

---

> > ### Comment · Reviewer_VjMs · 2025-04-05
> >
> > I would like to thank the authors for addressing my questions. The rebuttal adequately responded to the concerns raised in my review, and I will maintain my original decision.

---

> > > ### Author Response · Authors · 2025-04-05
> > >
> > > We would like to thank Reviewer VjMs for your insightful and detailed review, which allows us to further explore the generalizability of Transolver++. We will include all the newly added experiments and analyses in the future revision.
> > >
> > > Thanks for your dedication!

---

### Official Review · Reviewer_Ma1w · 2025-03-10

**Overall Recommendation:** 5

**Summary:**

Authors improve the scaling characteristics of a previously proposed model - Transolver - by analyzing computational and performative bottlenecks of the original model:

- homogeneous latent tokens (physical states) => inability to capture mesh details;
- memory bottleneck caused by processing large-scale meshes (~1M points) => inability to scale to large meshes,

which authors address by:
- computing projection weights by sampling the categorical distribution (Rep-Slice) with learnable temperature (Ada-Temp);
- distributing projection weights computation across multiple GPUs (as the operation is mainly point-wise).

**Claims And Evidence:**

Authors claim to achieve state-of-the-art performance on all the datasets, but on large geometries benchmarks (Table 2), the baseline range is quite limited. Mainly, there are graph neural network-based baselines, which are known to perform poorly on large meshes due to limited receptive field (mentioned by authors in the related works section). To support the claim, it would be beneficial to compare Transolver++ against more transformer-based architectures such as UPT (Alkin et al. 2024), GraphViT (Janny et al. 2023) and PointTransformer (Wu et al. 2024).

Another claim I find somewhat controversial is the high computational cost of traditional numerical methods. I know it is somewhat taken for granted within the community, but there are reports that the claim is based on rather imperfect baseline traditional solvers. I believe that a fair comparison with traditional solvers (given that they do not need dataset accumulation as neural-based approaches) would strengthen the claim.

**Essential References Not Discussed:**

As in the original Transolver paper, authors explore linear-time attention computed on latent tokens (physical states). This is very related to Universal Physics Transformers (Alkin et al. 2024), and I believe the comparison with UPT would improve the paper. Similarly, GraphViT (Janny et al. 2023) is another instance of attention over coarse-grained mesh representation which is not presented in the baselines. Additionally, linear-time transformers such as PointTransformer v3 (Wu et al. 2024) are not compared against.

**Experimental Designs Or Analyses:**

The experimental design is comprehensive and covers against-baselines comparisons as well as ablations and empirical scalability analysis.

**Methods And Evaluation Criteria:**

The benchmarks make perfect sense and are complete dataset-wise.

**Other Comments Or Suggestions:**

No suggestions other than including a couple more baselines mentioned above.

**Other Strengths And Weaknesses:**

Strengths:
- The paper is well written, the structure is easy to follow, and the notation is clear.
- I particularly appreciate the visual aspect of the paper and find the presentation impeccable and really well done.
- Authors focused on the scalability of the original paper and managed to improve it significantly by identifying key bottlenecks.
- The solutions to the issues are simple, elegant and scalable.
- Benchmarks are comprehensive, and the experimental design (up to a couple of additional baselines) is extensive.

Weaknesses:
- There are no error-bars in experiments. If that is possible to include them (at least for Transolver and Transolver++), that would be great.
- While the paper is a strong contribution application-wise, it is rather weak theoretically. My main concern is that the paper essentially improves a single model, but the analysis is strictly limited to that particular model. In a way, the paper is a significant engineering effort with next-to-perfect delivery, but the scientific contribution does not seem to be significant to me.
    - In particular, perhaps the strongest contribution of the paper is the slice reparameterization. It definitely works for Transolver, but I am not sure how much knowledge it adds to the community.
   - For comparison, next iterations of PointNet and PointTransformer significantly change respective architectures and are effectively different models derived from their predecessors. Transolver++ does not do that; if anything, it polishes the existing framework.

**Questions For Authors:**

Is it possible to include error bars for Transolver and Transolver++ in Table 2?

**Relation To Broader Scientific Literature:**

Key contributions are aligned with the overall trend for scalability in the DL community.

**Theoretical Claims:**

There are no theoretical claims except overhead analysis (Algorithm 1), and I believe it is correct.

---

> ### Author Rebuttal · Authors · 2025-04-01
>
> Many thanks to Reviewer Ma1w for your invaluable suggestions.
>
> >**Q1:** Lack of baseline range on large geometries benchmarks, such as UPT, GraphViT, and PointTransformer. Also, compare with imperfect baseline traditional solvers.
>
> **(1) New baselines.**
>
> Thank you for your constructive and helpful suggestions. Due to the limited rebuttal time, we were only able to carefully tune and compare several models on the **AirCraft** dataset, as summarized below. We will include these baselines along with brief model descriptions in the revised version. Notably, our model consistently achieves the best performance under this setting, demonstrating its effectiveness and robustness across strong baselines.
>
> |Model|Cp↓|$R_L^2$↑|Surf↓|
> |-|-|-|-|
> |UPT (Alkin et al., NeurIPS 2024)|0.035|0.994|0.112|
> |GraphViT (Janny et al., ICLR 2023)|0.041|0.990|0.130|
> |Point Transformer v3 (Wu et al., CVPR 2024)|0.045|0.987|0.145|
> |Transolver++|**0.014**|**0.999**|**0.064**|
>
> **(2) Imperfect baseline traditional solvers**
>
> For completeness, we also used a different traditional PDE solver (UnsCFD) to re-simulate the AirCraft data (smallest mesh scale in all three datasets). Each case took 5–6 hours on average, requiring 3–4 days to complete the full test set. In contrast, our neural model trains in under 10 hours and infers each case in less than 1 second, offering a significant efficiency advantage.
>
> Moreover, as shown below, **Transolver++ significantly outperforms UnsCFD in accuracy**, showing its potential as a practical surrogate for large-scale simulations. These results will be included in the appendix of the updated version.
>
> |AirCraft|Surf|
> |-|-|
> |UnsCFD|0.173|
> |Transolver++|**0.064**|
>
> > **Q2:** No error bars in Table2.
>
> Thank you for pointing this out, and we sincerely apologize for the oversight. All experiments for Transolver and Transolver++ were run at least three times. We provide the error bars (mean ± std) for Table 2 below and will include them in the revised version for clarity.
>
> |model|Volume|Surf|$C_D$|$R_L^2$|Surf|$C_l$|$R_L^2$|Surf|
> |-|-|-|-|-|-|-|-|-|
> |Transolver|0.173±0.003|0.167±0.002|0.061±0.002|0.931±0.005|0.145±0.008|0.037±0.003|0.994±0.001|0.092±0.003|
> |Transolver++|0.154±0.002|0.146±0.002|0.036±0.001|0.997±0.001|0.110±0.004|0.014±0.001|0.999±0.001|0.064±0.002|
>
> > **Q3:** Is it possible that the method improves a single model, but the analysis is strictly limited to that particular model? In particular, perhaps the strongest contribution of the paper is the slice reparameterization. I am not sure how much knowledge it adds to the community.
>
> Thanks for your detailed and instructive comments. As you mentioned, Transolver++ is specific to Transolver, which is why we name it "Transolver++," not others. However, the considered questions and design principles of our paper are valuable to this community. Here is the summarization.
>
> |Our design|Transferable Insights|
> |-|-|
> |local adaptive mechanism|PDE needs local-adaptive representations|
> |Parallelism framework|Parallelism is an effective way to large-scale tasks|
> |Industrial-level experiment|Encourage community to tackle large geometries|
>
> We would like to further clarify our contributions to this work. Our contributions lie in the following aspects.
>
> **1. A local adaptive mechanism to better learn the physical states.**
>
> The idea of **learning hidden physical states** is not specific to our model, which is widely shared in this community as a means of compression without explicit modeling. Our contribution lies in improving **how** such states are learned. The underlying principle is general and can be extended to other models with similar goals.
>
> **2. An end-to-end model with highly parallel implementation**
>
> To our best knowledge, Transolver++ is the first to validate end-to-end deep models and verify an effective parallelism framework for industrial datasets with million-scale geometries, which is **meaningful to the community**. Though tailored to Transolver, it provides insights into scalable architecture design and low overhead parallelism, which is ensuring the smallest communicating representation.
>
> It is also the first efficient parallelism framework in neural operators for PDE solving, which can serve as a starting point for exploring model-system co-design in solving PDEs. Thus, the contribution of the parallelism framework should not be underestimated.
>
> **3. Outstanding performance on industrial datasets**
>
> Our model consistently achieves SOTA results on both standard benchmarks and industrial datasets. As shown above, we further validated other models on the AirCraft dataset and our model on the newly-released DrivAerML (Ashton et al., CoRR 2024, 800 million mesh points per sample) datasets below.
>
> This outstanding performance firmly demonstrates the potential of advancing neural PDE solvers to real applications, which is a strong encouragement to the community.
>
> |Model|Cp↓|pPrime↓|
> |-|-|-|
> |Transolver|7.94|5.94|
> |Transolver++|**6.75**|**5.14**|

---

> > ### Comment · Reviewer_Ma1w · 2025-04-04
> >
> > I think the manuscript will be a valuable submission to the conference and, given new experiments, I have raised my score to highlight it. I appreciate authors' effort.

---

> > > ### Author Response · Authors · 2025-04-05
> > >
> > > We’re glad that our responses addressed your concerns, and we will carefully incorporate the corresponding revisions into the final manuscript, as outlined in the rebuttal. Thank you again for your support and for helping improve the quality of our work.

---

### Official Review · Reviewer_dnan · 2025-03-14

**Overall Recommendation:** 2

**Summary:**

This paper extends Transolver, which is a efficient transformer that predict the PDE solution of the input discretized geometry and physical quantities. The original Transolver weighted average the intermediate features at each grid node into several physical states tokens for efficient self-attention (physics-attention), and this weighted averaging can suffer from the over-smoothing problem. To address this issue, this paper propose to learn adaptive temperature at each node and use gumbel softmax trick. To further accelerate the computation of the physical states tokens, this paper also distribute the computation onto multi-gpus. To demonstrate the effectiveness of the proposed method, this paper conducted experiments on various synthetic benchmarks and real-world industrial applications, which proves that the two improvements introduced in this paper can handle mesh girds in higher resolution, which is essential to generate better results.

**Claims And Evidence:**

The claims made this paper is clear and convincing.

**Essential References Not Discussed:**

To the best of my knowledge, references are sufficient.

**Experimental Designs Or Analyses:**

The experimental designs and analysis are solid.

**Methods And Evaluation Criteria:**

The methods and evaluation criteria make sense in general, except for the adaptive temperature in Eq.3 is not guaranteed to be positive.

**Other Comments Or Suggestions:**

I do not have additional comments.

**Other Strengths And Weaknesses:**

- Strengths: This paper is well organized and generally well written.
- Weakness:
  - This paper contains some jargons make simple things complicated to understand: why it is called physical states while it is a weighted average of input features? __s__ is a weighted average of  __x__. They should have same physical interpretation, so it is confusing to name __s__ using additional term without any elaboration.  Although eventually this __s__ will be desliced and give the final output, it is really confusing to name __s__ through all layers as "state" since __s__ in early layers does not contain sufficient information that can be directly considered as a state of the unknown function to solve.
  - The improvement is so incremental that can hardly provide any new insights.
    - Oversmoothing is a well-known problem of attention, and the solution provided in the paper are a combination of existing techniques. Additionally, the authors did not mention how gumbel-softmax affect the deslice process during inference.
    - The acceleration in Sec.4.2 is also straightforward, additionally the __f__ is not explained anywhere in the paper.

**Questions For Authors:**

Please refer to the weakness. The major concern I have is that the improvement is so incremental and does not provide sufficient insights.
Therefore, I would expect authors could justify the contribution better and explain why the contributions are not incremental in the rebuttal.

**Relation To Broader Scientific Literature:**

The key contribution of this paper is leveraging previous techniques such as gumbel-softmax trick

**Theoretical Claims:**

This paper is about practical implementation improvements, there is no theoretical claim made.

---

> ### Author Rebuttal · Authors · 2025-04-01
>
> Many thanks to Reviewer dnan for providing a detailed and in-depth review.
>
> >**Q1:** Reclarification of the concept of "Physical States," particularly in early layers.
>
> Thank you for your rigorous review.
>
> **(1) "Physical states" is from Transolver (Wu et al., 2024)**
>
> In Transolver, "physical states" are defined as **physical internal-consistent representations** associated with specific geometric regions (e.g., windshield or sunroof of a driving car). Extensive visualizations in their work demonstrate its ability to learn such states. Besides, after confirming with its authors that **all the state visualizations in Transolver are based on the first layer representation**, we think this concept is reasonable for early layers.
>
> Since Transolver++ is an upgraded version of Transolver, we would like to keep this concept to ensure consistency.
>
> **(2) New experiments**
>
> To address your concern, in DrivAerNet++, we randomly sample 100 points across various regions and calculate JS divergence between **first-layer slice weights** among different areas. We can find that the slice weights present clear internal consistency and external inconsistency in different regions.
>
> Thus, slicing operation is not a trivial weighted sum. It can ensure the model to capture physical internal-consistent representations.
>
> |JS divergence|Sunroof-Part2|Windshield|Front|
> |-|-|-|-|
> |w.r.t. Sunroof-Part1|0.041|0.576|0.378|
>
> **(3) Representation learning perspective**
>
> Note that all model parameters will be simultaneously updated during training. In Transformer-based models, residual connections blur the distinction between early and ending layers, as gradients from the final loss can directly affect the first layer. Therefore, we cannot simply assume that early-layer representations are less related to physics.
>
> Considering visualizations of Transolver and concept consistency, we think "physical state" an appropriate term. It also clarifies our paper's core idea in "learning distinguishable states".
>
> >**Q2:** Reclarification of the novelty and significance of our work.
>
> To our best knowledge, Transolver++ is the first to design and validate deep models on million-scale industrial datasets and to verify an effective parallelism framework for PDEs, which is **meaningful to this community** in paving the way to practical neural solvers for real-world applications.
>
> Our technical contributions are summarized as follows.
> |Our design|New insights|
> |-|-|
> |local adaptive mechanism|PDE-solving requires local-adaptive representations|
> |Parallelism framework|Parallelism is an effective way to large-scale geometries|
> |Industrial-level experiment|Encourage the community to tackle large geometries|
>
> **(1) Local adaptive mechanism is better in learning physical states**
>
> Beyond only considering oversmoothing, our approach tries to enable local-adaptive for each point, where learned distributions are expected to be uniform for high-uncertainty areas and peaky for high-confident areas.
>
> To further demonstrate that our design is more than avoiding oversmoothing, we also provide ablation of "global annealing" temperature (a well-established way for oversmoothing attention). Results below confirm the effectiveness of our method. This also delivers the insight that local-adaptive representations are crucial in PDE solving.
>
> | AirCraft|$C_p$|$R_L$|Surf|
> |-|-|-|-|
> |Transolver + global annealing|0.034|0.993|0.093|
> |Transolver + Ada-Temp|0.020|0.995|0.080|
> |Transolver++|**0.014**|**0.999**|**0.064**|
>
> **(2) Highly parallel framework to enable million-scale geometries**
>
> While Eq (5) seems straightforward, it stems from our in-depth insights for Transolver and PDE-solving tasks, where we comprehensively investigated which representation should be parallel computed or communicated. With these efforts, Transolver++ successfully achieves the lowest communication overhead and presents linear scalability.
>
> It is also the first efficient parallelism framework in neural operators for PDE solving, which serves as a starting point for exploring model-system co-design in solving PDEs. Thus, the contribution of the parallelism framework should not be underestimated.
>
> **(3) Outstanding performance on industrial tasks**
>
> Our model achieves 13% and 20% promotion on standard benchmarks and industrial tasks. The new AirCraft dataset will also be made public to facilitate future research. More results are in Q1 and Q3 of Reviewer Ma1w.
>
> >**Q3:** How does Gumbel-softmax affect the deslice process?
>
> As shown in Eq.(5) and Algo 1, we generate new slice weights using Gumbel-Softmax rather than sampling a specific state, which allows the generated weights to be directly used for deslice.
>
> >**Q4:** f is not explained anywhere in the paper?
>
> As stated in Lines 267-270, we have stated that "Also, we found that ..., where x is used to generate slice weights, and f is combined with weights to form physical states." f represents geometric feature. We will highlight this in the revision.

---

### Decision · Program_Chairs · 2025-05-01

**Decision:**

Accept (poster)

**Comment:**

**(a) Summary**
This paper presents Transolver++, an extension of the original Transolver neural PDE solver, designed to scale to million-point mesh geometries via two key innovations: (1) a local adaptive mechanism using point-wise learned temperatures with Gumbel-Softmax to improve representation of physical states, and (2) an optimized multi-GPU parallelism framework that enables linear scalability and low communication overhead. The paper provides extensive experiments demonstrating significant performance gains on both standard PDE benchmarks and new industrial-scale datasets.

**(b) Strengths**
The reviewers generally agree that Transolver++ effectively tackles scalability challenges in neural PDE solvers and demonstrates state-of-the-art performance. Reviewer Ma1w highlighted the paper as "a valuable submission" and praised the authors for addressing bottlenecks with “simple, elegant and scalable” solutions. Reviewer VjMs emphasized the paper’s practical significance for large-scale scientific computing and acknowledged the empirical rigor and relevance of the proposed improvements. Reviewer dnan noted that the claims are convincing, and the experimental setup is solid. Despite the lack of theoretical novelty, the paper was seen as a significant engineering contribution by most reviewers.

**(c) Weaknesses**
Several concerns were raised:
* Reviewer oAjG and Reviewer dnan criticized the work for lacking conceptual clarity. Reviewer oAjG questioned the motivation for the adaptive temperature mechanism and found the terminology and methodology insufficiently explained, suggesting the solution may be unprincipled and model-specific.
* Reviewer dnan felt the contribution was incremental and not insightful, while also pointing out minor notational and explanation gaps.
* Reviewer Ma1w and Reviewer VjMs raised concerns about the limited range of baselines on large-scale benchmarks. Reviewer Ma1w suggested including comparisons to models like UPT, GraphViT, and PointTransformer. Both Ma1w and VjMs also noted the lack of theoretical depth, suggesting the work is mostly a well-executed extension rather than a groundbreaking methodological innovation.

**(d) Decision: Weak accept**
Overall, despite some disagreements about novelty and generality, the consensus favors acceptance due to the paper’s practical contributions, strong empirical results, and successful rebuttal. The authors provided additional baselines, clarified terminology and goals, and validated the adaptive and parallel mechanisms through ablations and new datasets. Reviewer Ma1w increased their score and expressed appreciation for the thorough responses, while Reviewer VjMs maintained support after the rebuttal. Reviewer oAjG acknowledged the rebuttal but remained skeptical. Reviewer dnan also acknowledged the authors' clarifications without updating their score. Given the strength of the practical contributions and the compelling experimental support, I recommend acceptance.

**Additional Comments on Reviewer Discussion**
Ma1w and VjMs were actively engaged in the rebuttal and updated or confirmed their positive assessments. dnan and oAjG acknowledged the rebuttal but did not fully engage or update their reviews. The authors made commendable efforts to clarify misunderstandings and add new experiments, which were appreciated by the more responsive reviewers.